# Enrichment of trace metals from acid sulphate soils in sediments of the Kvarken Archipelago, eastern Gulf of Bothnia, Baltic Sea

Joonas J. Virtasalo[1], Peter Österholm[2], Aarno T. Kotilainen[1], Mats E. Åström[3]

[1]Marine Geology, Geological Survey of Finland (GTK), Espoo, 02150, Finland
[2]Department of Geology and Mineralogy, Åbo Akademi University, Turku, 20500, Finland
[3]Department of Biology and Environmental Science, Linnaeus University, Kalmar, 39182, Sweden

Correspondence to: Joonas J. Virtasalo (joonas.virtasalo@gtk.fi)

**Abstract.** Rivers draining the acid sulphate soils of western Finland are known to deliver large amounts of trace metals with detrimental environmental consequences to the recipient estuaries in the eastern Gulf of Bothnia, northern Baltic Sea. However, the distribution of these metals in the coastal sea area, and the relevant metal transport mechanisms have been less studied. This study investigates the spatial and temporal distribution of metals in sediments at 9 sites in the Kvarken Archipelago, which is the recipient system of Laihianjoki and Sulvanjoki rivers that are impacted by acid sulphate soils. The contents of Cd, Co, Cu, La, Mn, Ni and Zn increase in the cores during the 1960s and 1970s as a consequence of intensive artificial drainage of the acid sulphate soil landscape. Metal deposition has remained at high levels since the 1980s. The metal enrichment in seafloor sediments is currently visible at least 25 km seaward from the river mouths. Comparison to sediment quality guidelines shows that the metal contents are very likely to cause detrimental effects on marine biota more than 12 km out from the river mouths. The dynamic sedimentary environment of the shallow archipelago makes these sediments potential future sources of metals to the ecosystem. Finally, the strong association of metals and nutrients in the same sediment grain size class of 2–6 µm suggests that the transformation of dissolved organic matter and metals to metal-organic aggregates at the river mouths is the key mechanism of seaward trace metal transport, in addition to co-precipitation with Mn-oxyhydroxides identified in previous studies. The large share of terrestrial organic carbon of the total organic C in these sediments (interquartile range = 39–48%) highlights the importance of riverine organic matter supply. These findings are important for the estimation of environmental risks and the management of biologically-sensitive coastal sea ecosystems.

## 1 Introduction

Acid sulphate (AS) soils are regarded as the nastiest soils in the world due to their ability to generate sulphuric acid and extremely low pH conditions in pore- and surface-waters (Dent and Pons, 1995). The global distribution of these soils is estimated at 50 million ha including areas in Australia, Africa, Central and South America, South and Southeast Asia, and Northern and Western Europe (Andriesse and van Mensvoort, 2006; Michael et al., 2017). In Europe, the largest occurrences of AS soils are probably found in Finland where current estimates point to AS soils occupying an area in the order of 1 million ha (Anton Boman, personal communication).

AS soils in northern Europe are organic-rich sulphide-bearing muds that were originally deposited in the Baltic Sea during its brackish-water phase, which, in the Gulf of Bothnia, began ca. 7000 years ago (Virtasalo et al., 2007; Häusler et al., 2017). These muds have since emerged above sea level as a result of rapid land uplift (today 4–9 mm yr$^{-1}$ in western Finland; Mäkinen and Saaranen, 1998; Kakkuri, 2012). Artificial drainage and reclamation of these lands for farming purposes, which was particularly intensive in Finland in the 1960s and 1970s, has caused a significant lowering of groundwater level (Saarinen et al., 2010; Yu et al., 2015). The groundwater lowering has enabled rapid oxidation of metal sulphide minerals that are abundant in these muds, producing $H_2SO_4$ and resulting in AS soils with a pH <4 (Yli-Halla et al., 1999; Sohlenius and Öborn, 2004; Boman et al., 2010). Under these extremely acidic conditions, large quantities of metals are released to the porewater due to

the oxidative dissolution of metal sulphides and weathering of silicate minerals. Particularly during high water flow conditions in spring and autumn, acidic porewater rich in metals (e.g. Al, Cd, Co, La, Mn, Ni and Zn) is flushed to recipient streams, with detrimental ecological consequences for biodiversity and the community structure of fish, benthic invertebrates and aquatic plants (Hudd and Kjellman, 2002; Fältmarsch et al., 2008; Sutela and Vehanen, 2017). Climate change is predicted to result in increasing precipitation and river discharges during winter in Finland, and increasing temperatures and evapotranspiration during summer (Olsson et al., 2015) that is likely to enhance drying and oxidation of sulphides in AS soils (Österholm and Åström, 2008; Job et al., 2020). It is thus expected that the acidic runoff and metal loading from AS soils will increase with climate change (Saarinen et al., 2010; Nystrand et al., 2016).

When acidic metal-rich river waters from the boreal AS soil landscape are discharged to estuaries with higher pH and salinity, the metals are complexed with organic matter or co-precipitated with Al-, Fe- and Mn-oxyhydroxides and consequently deposited in sediments (Åstöm and Corin, 2000; Nystrand et al., 2016). Low-density organic aggregates have the capacity to be transported across estuarine gradients, and their role in the seaward trace metal transport has been highlighted in recent studies, particularly in the Baltic Sea (Gustafsson et al., 2000; Jokinen et al., 2020). It has been shown that trace metals from AS soils are enriched in seafloor sediments near the mouths of rivers draining from the coastal plains of western Finland compared to background values and to the parent AS soil material (Nordmyr et al., 2008a, 2008b), with documented deteriorative effects on local benthic invertebrate communities (Wallin et al., 2015). Less is known, however, about the distribution of these metals in sediments further out from the river mouths, although this information is important for the estimation of ecotoxicological effects and the management of biologically-sensitive coastal sea ecosystems (e.g. de Souza Machado et al., 2016). The distribution pattern of the metals is also informative about the seaward metal transport mechanisms.

Permanent sediment deposition in the eastern coastal Gulf of Bothnia is restricted to small patches due to shallow water depths and the openness of the area to dominant southwesterly winds (waves) (Kotilainen et al., 2012). This pattern of sediment deposition and erosion is in slow but constant change due to the uplift, which makes these recent sediment patches prone to remobilization and potential secondary sources of metals to the marine ecosystem. In addition, the building and maintenance of offshore infrastructure such as shipping lanes requires repeated and, in the long term, more extensive dredging, which can result in resuspension and redistribution of metals in the environment (Lehoux et al., 2020). Finally, climate models project stronger westerly winds and shorter ice season in winter over Northern Europe (Ruosteenoja et al., 2019), which have the potential to increase seafloor erosion and sediment redistribution in the study area. Therefore, understanding the transport and distribution of metals in seafloor sediments is particularly important in the dynamic eastern coastal Gulf of Bothnia.

The aim of this study is to assess for the first time the distribution of trace metals in seafloor sediments with distance from the mouths of rivers running through a boreal AS soil landscape. The studied sea area is in the Kvarken Archipelago, off the town of Vaasa, in the eastern Gulf of Bothnia, which receives significant amounts of metals from the Laihianjoki and Sulvanjoki rivers that are among the most AS soil impacted rivers in Finland and Europe (Roos and Åström, 2005). The focus is on sediments deposited before and after the intensive artificial drainage of the AS soil landscape beginning in the 1960s. The focus is further on metals that are known to be extensively leached from AS soils (Al, Cd, Co, Cu, La, Mn, Ni and Zn) and much less so from the local industry and other human activities in the area (c.f. Åström and Björklund, 1997; Åström and Corin, 2000; Österholm and Åström, 2002).

## 2 Study area

The Kvarken Archipelago is located in the Gulf of Bothnia, where it marks the border between the Bothnian Sea in the south and the Bothnian Bay in the north (Fig. 1). The area was covered by the Fennoscandian Ice Sheet during the latest (Weichselian) glaciation, which retreated from the area ca. 10 400 years ago (Sauramo, 1929; Saarnisto and Saarinen, 2001; Stroeven et al., 2016). Kvarken is a UNESCO World Heritage Site because it is considered as a prime location demonstrating the effects of rapid glacioisostatic uplift (today ca. 8 mm yr$^{-1}$; Mäkinen & Saaranen, 1998; Kakkuri, 2012) on shoreline displacement and changes in coastal landscape. During and just after deglaciation, the archipelago was submerged to a water depth of 250–280 m, whereas today the area is very shallow (<25 m) and shoaly, with approximately 7000 islands and islets (Breilin et al., 2005; Ojala et al., 2013). The rapid uplift has led to strong seafloor erosion and sediment transport to deep areas further offshore. Till covers ca. 70 % of the modern seafloor in the archipelago (Kotilainen et al., 2012). Glaciolacustrine varved silts and clays, and postglacial lacustrine weakly-layered silty clays cover ca. 18 % of the seafloor, whereas bedrock outcrops are rare (3 % of the seafloor). Patches of recent mud deposition cover only ca. 8 % of the seafloor (Kotilainen et al., 2012).

The Kvarken Archipelago belongs to the continental subarctic climate zone with severe dry winters and almost warm summers. The mean annual air temperature is 4.2 °C, with the mean minimum temperature of 2.1 °C and the mean maximum temperature of 6.6 °C during the period 1981–2010 (Pirinen et al., 2012). The mean annual precipitation is 497 mm. The Bothnian Sea freezes on an annual basis and remains frozen for up to 140–150 days per year. The annual mean sea surface salinity in the archipelago ranges between 3.5 and 4, and the annual mean sea surface temperature between 3.5 and 7 °C. The sea is essentially non-tidal, but irregular water level fluctuations of as much as ±1.5 m take place as a result of variations in wind and atmospheric pressure. Stratification of the shallow waters is governed by a thermocline that develops each summer. The area is generally less affected by eutrophication and the associated seafloor oxygen deficiency, which are widespread in the southern and central Baltic Sea (Lundberg et al., 2009).

## 3 Materials and Methods

### 3.1 Sediment coring

The fieldwork was carried out during the summers of 2016–2018 onboard the research vessel *Geomari* of the Geological Survey of Finland. *Geomari* is equipped with a marine geological seismoacoustic survey system, which includes Meridata 28 kHz pinger and Massa TR-61A 3.5–8 kHz CHIRP sub-bottom profilers that were essential for the identification of coring sites that are representative of recent sediment deposition.

Altogether 9 sediment cores (Table 1) were collected using a Gemax twin-barrel short gravity corer (core diameter 9 cm), which preserves the soft sediment surface essentially undisturbed. One core of each twin was cut in half lengthwise and cleaned for sedimentological description and photography, whereas the other core was sectioned using a rotary device into 1 cm sample slices. The sample slices were stored in the cold (4–6 °C) and dark until shore-based laboratory analysis, which took place within a few months of the sample collection.

### 3.2 Laboratory analyses

Sample slices were analysed for [137]Cs activity content in order to constrain sediment chronology in each core. The [137]Cs activity of untreated samples was measured for 60 min using a BrightSpec bMCA-USB pulse height analyser coupled to a well-type NaI(Tl) detector at the Geological Survey of Finland (Ojala et al., 2017). Each core was analyzed starting from the uppermost sample slice and progressing downward until near zero (background) activity levels were measured in at least three consecutive samples. No corrections were applied for the results because the aim was only to detect relative [137]Cs activity peaks. Due to

the possible post-depositional downward transport of $^{137}$Cs through bioturbation and diffusion (Holby and Evans, 1996; Klaminder et al., 2012) the depth of peak $^{137}$Cs activity (rather than the initial increase) was assumed to represent the fallout from the 1986 Chernobyl nuclear disaster. Sample slices for each core were classified to those deposited in 1986 and later, and those deposited before 1986. The samples deposited before 1986 were further classified as those deposited after and before the year 1960, by calculating the average thickness of sediment deposited annually after 1986 in each core, and estimating the depth of 1960 by assuming a constant sedimentation rate for that core. This approach potentially slightly overestimates the depth of the year 1960 because of the increasing sediment compaction with core depth. The approach should, therefore, be viewed as conservative to sediments deposited before the year 1960.

After the non-destructive $^{137}$Cs analysis, fresh sample slices were freeze-dried, homogenized and halved, with one half analysed for multielement composition and the other for grain size distribution at the commercial laboratory Eurofins Labtium Ltd (Kuopio, Finland). The material for multielement analysis was sieved through a 63 µm mesh, and 0.2 g of the passed-through fraction was digested in a four-acid mixture of hydrofluoric acid, perchloric acid, hydrochloric acid and nitric acid (USGS Methods T01 and T20). After evaporation of the acids at 160 °C, the resulting gel was dissolved in 1 M HNO$_3$, and analysed for element concentrations by inductively coupled plasma-mass spectrometry (ICP-MS) or inductively coupled plasma-optical emission spectrometry (ICP-OES). Ag, As, Bi, Cd, Ce, Dy, Er, Eu, Gd, Hf, Ho, La, Lu, Nb, Nd, Pr, Sb, Sm, Sn, Ta, Tb, Th, Tl, Tm, U, Yb were analysed by ICP-MS, whereas Al, Ba, Be, Ca, Co, Cr, Cu, Fe, K, Li, Mg, Mn, Mo, Na, Ni, P, Pb, Rb, S, Sc, Sr, Ti, V, Y, Zn, Zr were analysed by ICP-OES. Because HF dissolves silicate minerals, the digestion is considered as "near-total digestion" (Hall et al., 1996). The commercial sediment reference materials QCGBMS304-6, QCMESS-4, QCNIST8704, CO153B and in-house standards were used for assessing measurement accuracy. Element concentrations for all reference materials measured with each sample batch fell well within ±10 % of the certified values. Mercury was measured separately by HNO$_3$ leach of 0.2 g samples through thermal decomposition, amalgamation and atomic absorption spectrometry (US EPA Method 7473). Solid-phase contents of carbon and nitrogen in the samples were analyzed by thermal combustion elemental analysis (TCEA). The pools of inorganic C and N are negligible in this setting (Virtasalo et al., 2005; Jilbert et al., 2018), hence no decalcification was conducted and the total contents are considered equal to organic C and N. Experimental precision for each element based on the standard deviations of duplicate analyses of selected samples is provided in Supplement.

To quantify the proportions of terrestrial plant-derived (%OC$_{terr}$) and phytoplankton-derived organic matter in the C pool, a simple binary mixing model was applied for the molar N/C ratio, assuming end-member values of (N/C)$_{terr}$ = 0.04 and (N/C)$_{phyt}$ = 0.13 following Goñi et al. (2003) and Jilbert et al. (2018):

$$\%OC_{terr} = \frac{(N/C)_{sample} - (N/C)_{phyt}}{(N/C)_{terr} - (N/C)_{phyt}} \times 100 , \qquad (1)$$

The model integrates a variety of terrestrial organic-matter sources ranging from fresh vascular plant detritus to more degraded soil organic matter into a single end-member. This is practical since effectively all of the organic matter transported by rivers passes through the soil reservoir before entering the coastal zone, therefore representing a mixture of variably degraded material (Jokinen et al., 2018).

Grain size distribution was determined for selected freeze-dried samples by wet-sieving through 20 mm, 6.3 mm, 2 mm, 0.63 mm, 0.2 mm and 0.063 mm ISO 3110/1 test sieves. The samples were pretreated with excess H$_2$O$_2$ to remove organic matter prior to the analysis. The <63 µm size fraction was further analyzed down to 0.6 µm using a Micromeritics Sedigraph III 5120 Xray absorption sedimentation analyzer. The sieving results were merged with sedimentation data in Sedigraph software. Median grain size was calculated according to the geometric Folk and Ward (1957) graphical measures implemented in

GRADISTAT 4.0 software (Blott and Pye, 2001). Clay is defined as grains finer than 2 μm, whereas mud is clay and silt (<63 μm), and sand is 63 μm to 2 mm (Blott and Pye, 2012).

### 3.3 Statistical analysis

Element contents below detection were rounded to half the detection limits so that approximate values could be used in the analyses.

In order to explore relationships between elements in the produced multielement dataset, a robust compositional principal component analysis (PCA) after isometric logratio (ilr) transformation (Filzmoser et al., 2009) was carried out using the `robCompositions` 2.0.8 package in the R 3.5.1 software environment. The resultant loadings and scores were back-transformed to centered logratio (clr) space for meaningful visualization and interpretation in a compositional biplot (Filzmoser et al., 2018).

Relationships between elements and grain size classes were explored using partial least squares regression 2 (PLSR2; Tenenhaus, 1998) as implemented in the `plsdepot` 0.1.17 package in the R 3.5.1 software environment. The PLSR2 results were validated using hierarchical partitioning (Chevan and Sutherland, 1991) as implemented in the `hier.part` 1.0.4 package (Nally and Walsh, 2004) in the R.

## 4 Results

Multielement and grain size data produced in this study are available in PANGAEA (Virtasalo et al., 2020a, 2020b).

### 4.1 Core description

The sediment cores are composed of typical, soft organic-rich brackish-water mud that is depositing along the Finnish coast. The cores have brownish-grey oxidized surface layers that are 1–3 cm thick, below which the colour quickly changes to dark grey or black, implying a sharp gradient to reducing conditions (Supplement; Virtasalo et al., 2005). The sediments are bioturbated and mottled by small burrows. Thin beds with thicknesses on the centimetre scale range are visible in places, representing the remnants of the primary sedimentary structure. The thin bedding likely does not record seasonal changes in deposition, but episodic seafloor reworking by short-lived storm-triggered flows (Virtasalo et al., 2014).  No sign of significant erosion or gap in deposition was observed in any of the cores upon visual inspection.

The sediments are poorly sorted: the interquartile range (IQR) of geometric sorting statistic of all sample slices is 3.0–3.3 µm, with a median of 3.1 µm (Folk and Ward, 1957; Blott and Pye, 2001). The grain size distributions are rather uniform throughout the cores: the IQR of median grain size of all sample slices is 1.96–2.54 µm, with a median of 2.18 µm (Supplement). The sediments are classified as clayey silt according to (Blott and Pye, 2012) and silty clay according to soil taxonomy (Soil Survey Staff, 1999).

### 4.2 Vertical distribution

Peak [137]Cs activity is easily distinguishable in all the studied sediment cores, which permits the confident identification of the depth of the Chernobyl fallout year 1986 in each core (Fig. 2). The clearly defined activity peak in each core excludes significant sediment reworking and post-event migration of [137]Cs, and supports the estimation of the depth of the year 1960 by assuming a constant sedimentation rate.

Contents of metals Cd, Co, Cu, La, Ni and Zn generally begin to increase approximately at the depth of the year 1960 in the studied cores (Fig. 2). An exception is MGGN-2017-19 from the eastern Korshamnsfjärden, close to the rivers, where the metal contents are high and variable with no clear trend below the core depth of 16 cm (early 1980s), but show an upward increasing trend above this level. The metals reach particularly high contents in MGGN-2017-20 from Varisselkä at ca. 1986, after which they begin to decrease at that site. In all cores from Korshamnsfjärden (MGGN-2017-17, MGGN-2017-18, MGGN-2017-19), the increasing trends of Co, Ni, La and Zn continue overall to the core top, whereas the increasing trends of Cu and Cd level out or turn to decrease at ca. 1986. In cores from farther out at sea in Gloppet (MGGN-2018-29, MGGN-2018-31) from ca. 1986 to the core tops, the contents of Co, Ni and La vary at high values, whereas Zn, Cu and Cd show a decreasing trend.

Aluminium contents are variable but show no clear trends in the cores, except in MGGN-2017-20 from Varisselkä, where the vertical distribution of Al is similar to that of other metals in that core (Fig. 2).

Manganese contents are high and variable with an upward increasing trend in cores from Varisselkä, and eastern and middle Korshamnsfjärden (MGGN-2017-18, MGGN-2017-19, MGGN-2017-20; Fig. 2). In cores from farther out at sea in western Korshamnsfjärden and Gloppet (MGGN-2017-17, MGGN-2018-29, MGGN-2018-31), Mn contents are low except for a strong increase at the core tops.

Carbon contents increase toward the top in all cores. In cores from Gloppet (MGGN-2018-29, MGGN-2018-31) and from western Korshamnsfjärden (MGGN-2017-17), C contents range between 2 and 3 % prior to ca. 1986, and between 3 and 4 % higher up the cores (Fig. 2). In middle and eastern Korshamnsfjärden (MGGN-2017-18, MGGN-2017-19), C contents vary between 3 and 4 % before ca. 1986, and between 4 and 5 % in the upper core sections. In Varisselkä (MGGN-2017-20), C contents increase strongly between 1960 and 1986, and reach 6 % in the upper section of that core. The terrestrial organic share of the C content before ca. 1986 is 40–50 % in cores from Gloppet and western and middle Korshamnsfjärden, and 50–60 % in cores form eastern Korshamnsfjärden and Varisselkä. After ca. 1986, the share of terrestrial organic carbon decreases in all cores, largely mirroring the upward-increasing C content.

### 4.3 Statistical relationships

Statistical analyses were carried out on the upper core sections that were deposited after the year 1960 because it is clear in the vertical metal content profiles (Fig. 2) that this interval is the most enriched in metals.

The first principal component (PC1) of the robust compositional PCA explains 73.7 % of the total variance. PC2 explains 10.7 %, whereas the rest of the components each explain less than 6 % of the total variance. The metals Co, Ni and Cd cluster along the positive side of PC1 in the biplot (Fig. 3). Also Mn has a strong positive loading on PC1, but it deviates slightly from the other metals. This deviation of Mn is likely explained by its vertical distribution in sediment cores that is similar to the other metals in Varisselkä and in eastern and middle Korshamnsfjärden, but different in cores collected farther out at sea (Fig. 2).

A two component PLSR2 model utilizes 82.3 % of the variance of predictor variables (54 elements) to explain 68.3 % of the variance of response variables (13 grain size classes). The metals Cd, Co, Cu, Ni and Zn have strong positive correlations with the grain size classes of 2–4 and 4–6 µm in the PLSR2 (Fig. 4a). Notably, also C and N are positively correlated with these grain size classes.

In concordance with the PLSR2, the hierarchical partitioning analysis shows that the 2–4, 4–6 and 1–2 µm classes have the most independent power among grain size classes in predicting Ni contents, and account for 17.8, 13.1 and 11.8 % of the explained variance, respectively (Fig. 4b). The hierarchical partitioning patterns of the other elements identified in the PLSR2 are similar.

**4.4 Spatial distribution**

Metal contents are compared between core sections deposited before the year 1960 and those deposited in 1986 and later in order to explore the magnitude of recent metal enrichment (Fig. 5). Metal contents decrease toward the bottom in all cores; however, in Varisselkä, and Korshamnsfjärden East and Middle, metals do not decrease to low levels comparable to those in western Korshamnsfjärden and Gloppet (Fig. 2). Therefore, mean median metal contents in the pre-1960 sections of the four cores from Gloppet (MGGN-2018-29, MGGN-2018-30, MGGN-2018-31 and MGGN-2018-32) are considered to be representative of the local background values, and are used here as reference values in the depiction of spatial trends.

Median contents of Cd, Co, Cu, La, Mn, Ni and Zn in sediment cores decrease with distance from the Laihianjoki and Sulvanjoki rivers in the east (Fig. 5; Table 2; Supplement). Comparison between core sections deposited before 1960 and in 1986 and later shows that median contents of Cd, Co, Cu, La, Ni and Zn are higher in the upper sections of all cores, with the difference increasing toward east. However, Mn median contents are enriched only in the upper sections of the four cores closest to the rivers (MGGN-2016-8, MGGN-2017-18, MGGN-2017-19, MGGN-2017-20), whereas in the cores further offshore, there is essentially no difference between the upper and lower core sections (Fig. 5). In contrast, median Al contents are essentially uniform in all the cores, and between upper and lower sections.

When sediment cores are arranged according to distance from the Laihianjoki and Sulvanjoki rivers, the patter of decreasing metal contents with increasing distance is evident (Fig. 6). The median metal contents also exceed several sediment quality guideline thresholds. The metal contents have not been normalized because the sediment samples contain more than 30 % clay, and on median 3.7 % C, which means that normalization coefficients according to the Finnish sediment dredging and dumping guidelines are close to 1.

**5 Discussion**

Contents of trace metals known to be abundantly leached from AS soils (Al, Cd, Co, Cu, La, Mn, Ni and Zn) have been studied in sediment cores from the Kvarken Archipelago, which is the recipient sea area of the Laihianjoki and Sulvanjoki rivers. These rivers are frequently heavily loaded with a range of metals that certainly are derived from AS soils that are widespread in their catchment areas, whereas there are no other significant metal-releasing activities such as old or current mines or metal industry (Roos and Åström, 2005).

**5.1 Metal distribution**

The median contents of Al are similar in the core sections deposited before the year 1960 and in 1986 and later, and the values show essentially no change with distance from the river mouths (Fig. 5). The vertical distribution of Al is similar to the other metals with the highest values in the 1980s in the core from Varisselkä (MGGN-2017-20), but different at the other sites (Fig. 2). Clearly, the intensive artificial drainage of the AS soil landscape, which began in the 1960s (Saarinen et al., 2010; Yu et al., 2015), has not substantially influenced the delivery of Al to the coring sites other than Varisselkä. IQR of Al contents in sections deposited in 1986 and later in all cores is 67900–75200 mg kg$^{-1}$ (Table 2), and the maximum Al content is 110 000 mg kg$^{-1}$. A higher median Al content (86400 mg kg$^{-1}$; Table 2) has been reported near the mouth of the Vöyrinjoki river

(Nordmyr et al., 2008b), which is situated ca. 37 km northeast from the nearest coring site (Fig. 1b). This is in good agreement with previous observations, which show that Al to a large extent is deposited very close river mouths together with organic material (Nordmyr et al., 2008a, 2008b; Åström et al., 2012; Nystrand et al., 2016). Wallin et al. (2015) report the Al content of 59900 mg kg$^{-1}$ for a single sample "from the first accumulation basin in the estuary" of the Laihianjoki river (Table 2), but do not provide coordinates or a map of the sampling location, which makes it difficult to assess the representativeness of the sample.

Manganese median contents are enriched at the four sites closest to the rivers in the east (MGGN-2016-8, MGGN-2017-18, MGGN-2017-19, MGGN-2017-20) compared to cores collected farther out at sea (Fig. 5). The enrichment is stronger in the upper core sections deposited in 1986 and later than in the core sections deposited before 1960. The enrichment is in line with the previously documented increase of metal loading to estuaries in western Finland due to increased artificial drainage of the AS soil landscape, beginning in the 1960s (Yu et al., 2015, 2016). The vertical distribution of Mn shows elevated values in upper core sections similar to the other metals at the easternmost sites (Fig. 2; MGGN-2017-18, MGGN-2017-19, MGGN-2017-20). However, in cores farther out at sea in western Korshamnsfjärden and Gloppet, the vertical distribution of Mn is generally flat except a pronounced increase at the core tops. IQR of Mn in core sections deposited in 1986 and later in the easternmost sites is 3740–7300 mg kg$^{-1}$, whereas it is 548–1293 mg kg$^{-1}$ at the sites farther out at sea. The maximum Mn content in the upper core sections is 14000 mg kg$^{-1}$. Similar Mn contents to the easternmost sites have been reported from the open sea areas of Bothnian Sea and Bothnian Bay: mean 3000 ±1600 mg kg$^{-1}$ and 8500 ±5300 mg kg$^{-1}$, respectively (Leivuori and Niemistö, 1995). Higher Mn contents (median 9013 mg kg$^{-1}$; Table 2) have been reported near the mouth of the Vöyrinjoki river (Nordmyr et al., 2008b). Manganese enrichment in upper core sections is evident from Varisselkä (MGGN-2017-18) to eastern and middle Korshamnsfjärden (MGGN-2017-19, MGGN-2017-20), which shows that it is transported longer distances from the rivers than Al. This observation is in line with previous studies, which demonstrate that Mn can travel a long distance before precipitation as Mn-oxyhydroxides and the consequent deposition on the seafloor (Nordmyr et al., 2008a, 2008b; Nystrand et al., 2016). Even farther out at sea, the strong increase of Mn at the core tops (Fig. 2) is due to the reductive dissolution of buried Mn-oxyhydroxides and associated release of Mn$^{2+}$ into the porewater with subsequent upward diffusion and oxidative precipitation of Mn as oxyhydroxides in the sediment surface layer (Widerlund and Ingri, 1996). Mn is the only metal from AS soils, for which such redox-driven migration has been previously observed in the area (Nordmyr et al., 2008b). The Mn content reported by Wallin et al. (2015) from the Laihianjoki estuary is comparably low (788 mg kg$^{-1}$; Table 2).

The median contents of Cd, Co, Cu, La, Ni and Zn are higher at the four easternmost sites closest to the rivers (MGGN-2016-8, MGGN-2017-18, MGGN-2017-19, MGGN-2017-20) compared to those farther offshore (Fig. 5; Supplement). In contrast to Al and Mn, these metals are enriched in the upper core sections deposited in 1986 and later at all the coring sites compared to the lower sections deposited before 1960. Vertical distributions of these metals show increasing upward trends beginning at ca. 1960 in all cores, except in MGGN-2017-19 (eastern Korshamnsfjärden), where the initial metal contents are high and variable with no clear trend until they begin to increase in the early 1980s (Fig. 2). IQRs of Cd, Ni and Zn, for example, in core sections deposited in 1986 and later are 0.75–1.40 mg kg$^{-1}$, 51–107 mg kg$^{-1}$, and 254–454 mg kg$^{-1}$, respectively (Table 2). Similar Cd contents have been reported from the Laihianjoki estuary (0.92 mg kg$^{-1}$; Wallin et al., 2015) and the open sea area of Bothnian Bay (mean 0.8 ±0.3 mg kg$^{-1}$; Leivuori and Niemistö, 1995), whereas lower Cd contents have been reported from the open Bothnian Sea (0.4 ±0.2 mg kg$^{-1}$; Leivuori and Niemistö, 1995). A higher Ni content has been reported from the Laihianjoki estuary (130.5 mg kg$^{-1}$; Wallin et al., 2015), and higher Zn contents from the estuaries of Laihianjoki (461 mg kg$^{-1}$; Wallin et al., 2015) and Vöyrinjoki (maximum 608.5 mg kg$^{-1}$; Nordmyr et al., 2008b). However, the maximum Cd, Ni and Zn contents in the upper core sections are generally 2–3 times higher than previously reported from the area: 3.11 mg kg$^{-1}$, 245 mg kg$^{-1}$, and 835 mg kg$^{-1}$, respectively.

After the strong increase in sedimentary metal contents during the 1960s and 1970s, the metal contents and thus metal loading from the AS soils has stayed overall at the same level since the 1980s (Fig. 2). In Korshamnsfjärden (MGGN-2017-17, MGGN-2017-18, MGGN-2017-19), the contents of Co, Ni, La and Zn generally continue to increase until the core top, whereas the increasing trends of Cu and Cd level out or turn to decrease at ca. 1986. In Gloppet (MGGN-2018-29, MGGN-2018-31), more than 25 km from the river mouths, the contents of Co, Ni and La vary at high values up to the core top, whereas Zn, Cu and Cd begin to decrease from ca. 1986 onwards. An exception to this pattern is Varisselkä (MGGN-2017-20), where unexpectedly high contents of Al, Cd, Co, Cu, La, Ni and Zn were deposited in the early to mid-1980s (Fig. 2). In this core, the metal contents decrease toward the core top; however, despite this Varisselkä still has higher metal contents in the sediment surface than the other sites. The decrease in metal contents in Varisselkä parallels the decreasing share of terrestrial organic carbon since the 1980s, which suggests that the decrease in metal contents may be due to the reduced transport of metal-organic aggregates to the site (Section 5.2) as a result of, for example, narrowing of the shallow channel to the southeast (Fig. 5), rather than a decrease in the metal loading to the archipelago. Had the metal loading to the sea area decreased, it would certainly be visible also in the Korshamnsfjärden cores, which it is not.

In AS soils worldwide, such as in Australia, Fe species are typically recognized as a major product of acidic drainage, and a major sink for mobilized metals (e.g. Bush et al., 2004; Mosley et al., 2018; Job et al., 2020). However, in boreal AS soils, the mobility of Fe is typically low (Österholm and Åström, 2002; Sohlenius and Öborn, 2004; Nordmyr et al., 2008b). The reason for this condition has not yet been fully established, but it is probably related to efficient oxidation of the mobile $Fe^{2+}$ to the relatively insoluble $Fe^{3+}$, and that once formed, the oxidised form is protected from re-reduction. Consequently, the iron released from iron-sulfide minerals is largely retained within, and thus to only a limited extent leached from, the boreal AS soils. For example, Åström and Björklund (1996) have demonstrated that in unfiltered water samples from a stream draining AS soils in the boreal zone, there is no increase in Fe concentrations as the relative proportion of this soil type increases downstream. Furthermore, and the majority of Fe transported by acidic rivers to the estuaries has been shown to precipitate and deposit close to river mouths (Nordmyr et al., 2008a; Åström et al., 2012; Nystrand et al., 2016). The seaward distribution pattern of Fe in the Kvarken Archipelago is comparable to that of Al; there is no Fe enrichment in the upper core sections deposited in 1986 and later (IQR 40300–45600 mg $kg^{-1}$; median 43200 mg $kg^{-1}$) compared to the lower sections deposited before 1960 (IQR 43300–47800 mg $kg^{-1}$; median 45600 mg $kg^{-1}$), and there is no systematic decrease in the Fe contents with distance from the river mouths (Supplement). Finally, the measured Fe contents in the sea area are not enriched compared to their parent AS soils with a median Fe content of 38000 mg $kg^{-1}$ (90[th] percentile 48100 mg $kg^{-1}$; Åström and Björklund, 1997).

It is worth noting that permanent sediment deposition is today restricted to small patches in the eastern coastal Gulf of Bothnia due to the shallow water depths and openness of the sea area to dominant southwesterly winds (waves) (Kotilainen et al., 2012). As a consequence, finding coring sites that are representative of the recent sediment deposition can be challenging in the area without a guidance from seismoacoustic sub-bottom surveys such as those carried out here. For example, metal contents reported by Wallin et al. (2015) are generally lower than those measured here, although their sampling site supposedly was closer to the source rivers. The four-acid digestion method used in this study generally produces comparable results for metals from AS soils to previous studies that have aimed at analysing "total metal contents" in sediments, although different methods were used (Cook et al., 1997; Nordmyr et al., 2008a, 2008b).

## 5.2 Metal transport mechanisms

The similar distribution of Cd, Co, Cu, La, Ni and Zn in the studied sea area is supported by the PCA, which shows similar behaviour of the metals, in particular Cd, Ni, and Co (Fig. 3). Previous studies of water samples, sediment trap material and

seafloor sediments have concluded that Cu and La precipitate readily close to river mouths, whereas Cd, Co, Ni and Zn are preferentially transported a bit further out where they most likely co-precipitate and are deposited with Mn-oxyhydroxides (Nordmyr et al., 2008a, 2008b; Nystrand et al., 2016). This study shows that Cd, Co, Cu, La, Ni and Zn all are enriched in sediment cores farther out at sea than Mn, which strongly indicates that other mechanism(s) in addition to the precipitation of Mn-oxyhydroxides influence their seaward transport and distribution.

Field studies and geochemical modelling show that Cd, Co, Cu, La, Ni and Zn in AS soil impacted rivers are transformed from dissolved to particulate form as they are discharged to the sea (Nordmyr et al., 2008a, 2008b; Nystrand et al., 2016). Seaward transport of suspended particles is highly dependent on hydrological conditions, with high discharge producing large plumes of river water by which metals can be transported far from the river mouths (Nystrand et al., 2016). Exceptionally large river plumes can be caused by extreme events such as that in the late autumn of 2006, when a severely dry summer (maximising oxidation of sulphides in the AS soils) was succeeded by a severe wet spell (Österholm and Åström, 2008; Saarinen et al., 2010), causing widespread fish kills in rivers and estuaries in western Finland. Some of the peaks observed in the metal vertical distributions in sediment cores (Fig. 2) may result from such extreme events; however, exceptional events hardly explain the overall metal enrichment in the cores.

The PLSR2 analysis, supported by hierarchical partitioning, shows that Cd, Co, Cu, Ni and Zn are strongly positively correlated with sediment grains of the size between 2 and 6 µm (Fig. 4). Also the nutrients C and N have strong positive correlations with the same grain-size range, which suggests that the metals are associated with organic particles. This observation is supported by recent studies, which demonstrate the importance of metal-organic matter aggregates in land-to-sea transfer of trace metals, particularly in boreal environments (Jokinen et al., 2020). When acidic river water is discharged to the sea, dissolved Al and Fe precipitate and are deposited as oxyhydroxides as a consequence of neutralization by mixing with seawater (Nordmyr et al., 2008a; Åström et al., 2012; Nystrand et al., 2016), whereas trace metals may behave more conservatively and form complexes with organic matter (Simpson et al., 2014) that is also transformed from dissolved to particulate form as a result of salinity-induced flocculation (Sholkovitz, 1976; Asmala et al., 2014). The produced low-density metal-organic aggregates can be transported by currents far from the river mouth, as has been demonstrated for the Kalix River in the Bothnian Bay (Gustafsson et al., 2000), and elsewhere (Wang et al., 2017; Pavoni et al., 2020a, 2020b).

Organic aggregates in coastal environments are loosely bound and fragile, and have a size range of tens to thousands of micrometers (Eisma, 1986; Mikkelsen et al., 2006; Lee et al., 2012). The aggregates are easily broken after deposition by benthic organisms (e.g. Rhoads and Boyer, 1982), by sediment compaction with burial and, ultimately, by the grain size analysis, to their constituent particles, which typically are smaller than 20 µm (Eisma, 1986; Mikkelsen et al., 2006; Lee et al., 2012). The 2–6 µm size range identified here differs from the local phytoplankton community, which is dominated by species smaller than 2 µm during summer, and those larger than 10 µm in winter and spring (Andersson et al., 1996; Paczkowska et al., 2017). Indeed, sample treatment with excess $H_2O_2$ prior to the grain size analysis largely leaves behind inorganic particles, such as mineral grains that are commonly found enclosed in organic aggregates (Eisma, 1986; Mikkelsen et al., 2006; Lee et al., 2012). The 2–6 µm size range is close to the size of lithic grains with a modal peak of 8 µm that are transported by the Karjaanjoki river to the Pojoviken estuary in the southern Finland, where the lithic grains are passively enclosed in organic aggregates at the river mouth and transported seaward (Joonas Virtasalo, personal communication). The 2–6 µm range is slightly larger than the median sediment grain size, the IQR of median grain size of the studied samples being 1.96–2.54 µm. The share of 2–6 µm grains is relatively invariable in each core, generally between 20–25 % (Supplement), which further indicates that the observed vertical metal enrichment patterns are not controlled by temporal changes in sediment transport, but by external metal loading such as that from AS soils.

The importance of riverborne organic matter in the sea area is demonstrated by the large share of terrestrial organic carbon in the cores. IQR of the share of terrestrial organic carbon of the total organic C in the cores is 39.2–47.8 %, which is substantially higher than in, e.g. coastal sea areas of southern Finland, where the terrestrial share usually is less than 30 % (Jilbert et al., 2018; Jokinen et al., 2018, 2020). The terrestrial share is highest in cores from eastern Korshamnsfjärden (MGGN-2017-19) and Varisselkä (MGGN-2017-20) closest to the river mouths (Fig. 2). The share of terrestrial organic carbon decreases upward in core sections deposited in 1986 and later, largely mirroring the increase in total C, which indicates that the increase in total organic C is largely driven by increasing phytoplankton production during the recent ca. 30 years, in line with the observed increasing nutrient levels in the inner Kvarken Archipelago since 1980 (Lundberg et al., 2009).

**5.3 Risk assessment**

Metals associated with particles are eventually settled and buried in sediments, and are therefore less available for the aquatic biota. However, particulate metals in sediments may be toxic to benthic invertebrates via gastrointestinal tract and skin (Eggleton and Thomas, 2004; Wallin et al., 2015). Metals may also be dissolved from sediments to the aqueous phase if seafloor physical-chemical conditions are altered or sediment is bioturbated (Eggleton and Thomas, 2004; de Souza Machado et al., 2016).

Finnish sediment dredging and dumping guidelines provide metal content thresholds for the assessment of the suitability of material for offshore dumping (Ympäristöministeriö, 2015). Level 2 thresholds in the guidelines are defined so that metal contents exceeding the levels cause acute toxicity in less than 5 % of marine organisms. Zinc contents in the majority of the samples and Cd contents in half of the samples that were deposited in 1986 and later in Varisselkä (MGGN-2017-20) exceed the level 2 threshold, which means that these sediments are considered unsuitable for offshore disposal (Fig. 6). Nickel contents in almost half of the samples exceed the level 2 threshold as far as 14.2 km from the nearest river mouth (MGGN-2017-17, western Korshamnsfjärden).

Finnish stakeholders often use North American and Canadian guidelines when assessing the environmental impacts of metals in marine sediments because of similar geological environment (e.g. Vallius, 2015). The North American guidelines determine metal toxicity in sediment relative to two threshold levels: effects range-low (ERL) and effects range-medium (ERM). Metal contents exceeding the ERMs frequently result in adverse effects on biota, whereas metal contents between ERLs and ERMs occasionally result in adverse effects on biota, and metal contents below ERLs rarely result in adverse effects on biota (Long et al., 1995). The Canadian Guidelines for Protection of Aquatic Life consist of the Interim Sediment Quality Guidelines (ISQGs) and the Probable Effect Levels (PELs), which are used to evaluate the biological effects of a contaminant (Canadian Council of Ministers of the Environment, 2001). Contents exceeding the PELs frequently result in adverse effects on biota, whereas levels between the PELs and the ISQGs are associated with infrequently occurring adverse effects. Levels below the ISQG rarely cause adverse effects.

More than half of the measured Zn contents exceed the ERL and ISQG levels at each coring site, frequently even in the core sections deposited before 1960 (Fig. 6). Furthermore, the majority of measured Zn contents exceed the ERM and PEL levels as far as 12.6 km from the nearest river mouth (MGGN-2017-18, middle Korshamnsfjärden). The majority of measured Ni contents exceed the ERL at each coring site, and the ERM as far as 14.2 km from the nearest river mouth (MGGN-2017-12, western Korshamnsfjärden). The majority of Cd contents exceed the ISQL as far as 24.7 km from the nearest river (MGGN-2018-30, Gloppet), and the ERL at 12.6 km from the nearest river (MGGN-2017-18, middle Korshamnsfjärden). It seems likely that metal loading from AS soils has detrimental effects on biota in the studied sea area. This simple assessment would

benefit from determining the speciation of metals in sediments, as it affects the toxicity (c.f. Linge, 2008). Furthermore, instead of individual metals, the combined toxic effects of several metals and environmental factors should be considered (Chu and Chow, 2002). The ecotoxicological risk of metal loading from AS soils was previously assessed to be high in the area by Wallin et al. (2015).

## 6 Conclusions

Loading from AS soils has resulted in the strong enrichment of Cd, Co, Cu, La, Mn, Ni and Zn in sediments of the Kvarken Archipelago. The loading intensified in the 1960s and 1970s, when previous studies show that intensive artificial drainage of the coastal AS soil landscape begun. Unlike the case for many AS soils worldwide, Fe is not enriched in the recipient sea area.

The metal deposition has remained at more or less the same level since the 1980s, however with fine-scale variability in
contents both among the metals and sampling sites. Metal transport from the Laihianjoki and Sulvanjoki rivers toward open sea largely takes place along Korshamnsfjärden. The metal enrichment in seafloor sediments is currently visible at more than 25 km distance from the rivers. Comparison to sediment quality guidelines shows that metal contents in the majority of analysed sub-samples are sufficiently high to very likely have detrimental effects on marine biota more than 12 km from the river mouths. The dynamic nature of the patchy sediment deposition, the rapid uplift of the region, and the predicted increase
in storm wave erosion with climate change imply that these sediments are potential future sources of metals to the marine ecosystem. Furthermore, acidic runoff and metal loading from acid sulphate soils have been predicted to increase with the climate change.

Previous studies have identified Mn-oxyhydroxides as a mechanism of metal transport and deposition seaward from the river
mouths in the area. This study shows that Cd, Co, Cu, La, Ni and Zn are transported further out at sea than Mn, which requires an additional mechanism of metal transport. The strong association of the metals and nutrients to sediment grains of the same size range ($2–6\,\mu m$) suggests that the transformation of dissolved organic matter and metals to metal-organic aggregates at the river mouths is the key mechanism of seaward trace metal transport. The large share of terrestrial organic carbon of the total organic C in these sediments (interquartile range 39–48 %) highlights the importance of riverine organic matter supply.

*Data availability.* Multielement and grain size data are available in PANGAEA (Virtasalo et al., 2020a, 2020b).

*Supplement.* The supplement related to this article is available online at: [journal web page]

*Author contributions.* Joonas Virtasalo: Writing – original draft, Conceptualization, Investigation, Formal analysis, Funding acquisition. Peter Österholm: Writing – review and editing, Conceptualization. Aarno Kotilainen: Writing – review & editing, Investigation, Funding acquisition, Project administration. Mats Åström: Writing – review and editing, Conceptualization.

*Competing interests.* The authors declare that they have no conflict of interest.

*Acknowledgements.* This study resulted from the SmartSea project, funded by the Strategic Research Council at the Academy of Finland (grant number 292 985). M.E.Å. additionally acknowledges the Swedish Research Council Formas (grant number 2018-00760). The study has utilized research infrastructure facilities provided by FINMARI (Finnish Marine Research

Infrastructure network). Help by the crew of R/V *Geomari* and those who assisted in the field work is gratefully acknowledged. Satu Vuoriainen carried out the [137]Cs analyses. The reviewers are thanked for comments that helped improve the manuscript.

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

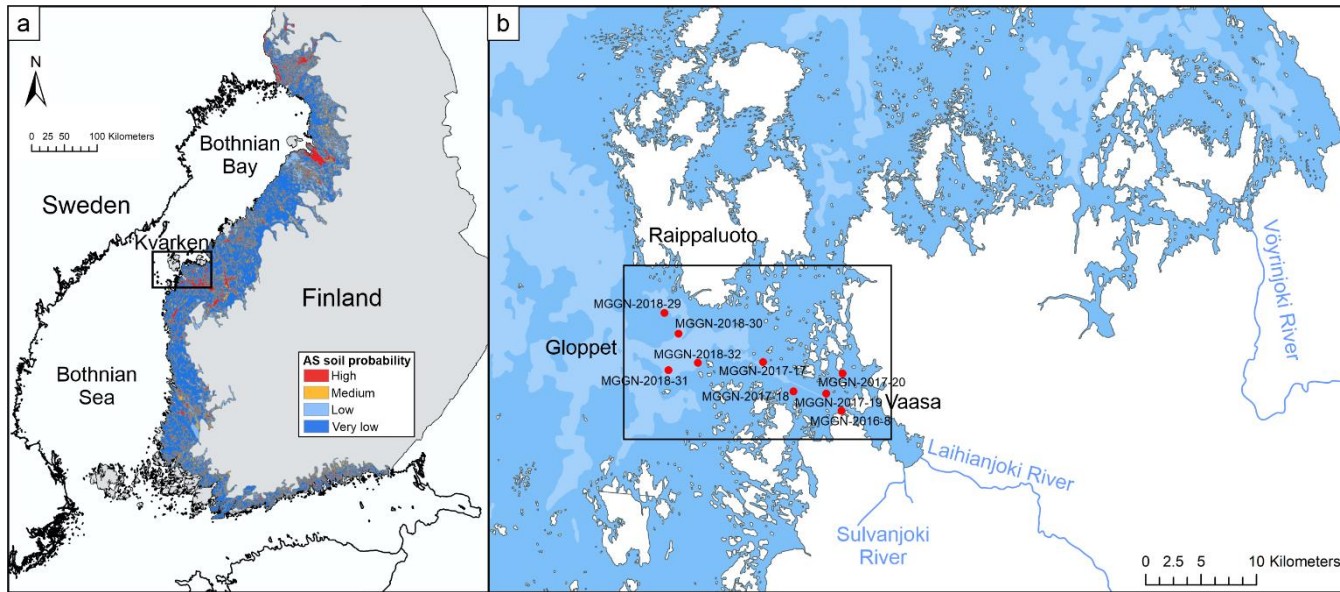

**Figure 1. Maps of the Baltic Sea and study area. (a) Map of the Baltic Sea, and probability of acid sulphate soil occurrence in Finland. Black square indicates the location of the study area on the west coast of Finland. (b) Nautical chart of the study area in the Kvarken Archipelago. Red dots indicate the sediment coring sites of this study. Acid sulphate soil probability map: Geological Survey of Finland 2018. Nautical chart: S-57 Finnish Transport Agency 2017.**


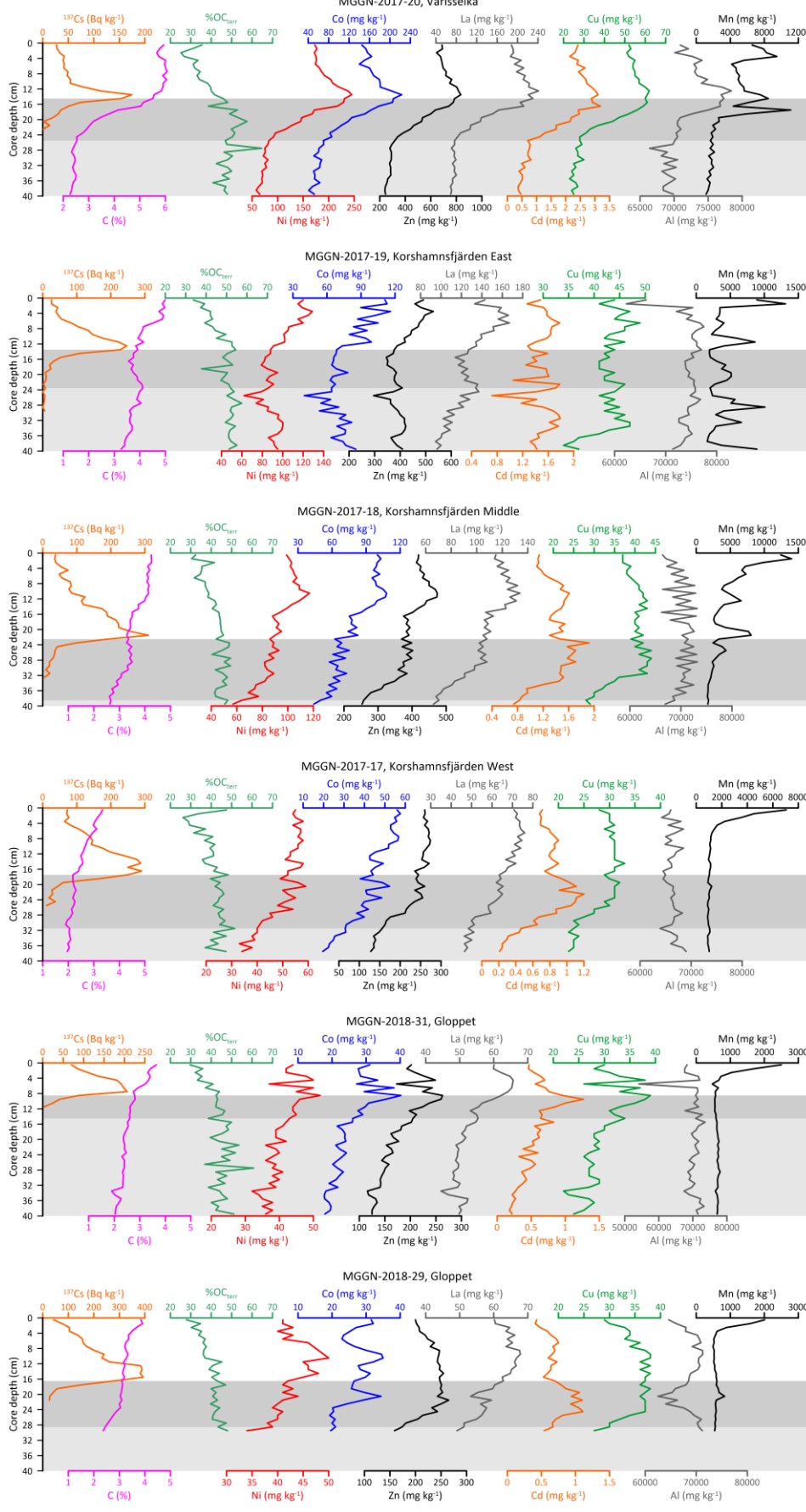

**Figure 2. Vertical distributions of $^{137}$Cs, carbon, the share of terrestrial organic carbon, and metals typically leached from acid sulphate soils in the <63 µm grain size fraction of representative sediment cores in the Kvarken Archipelago. The cores are arranged according to increasing distance from the Laihianjoki and Sulvanjoki rivers (downward). The dark grey shading indicates the interval deposited between 1960 and 1985, and the light grey shading indicates the interval deposited before the year 1960 in each core. The intervals without shading were deposited in 1986 and later.**


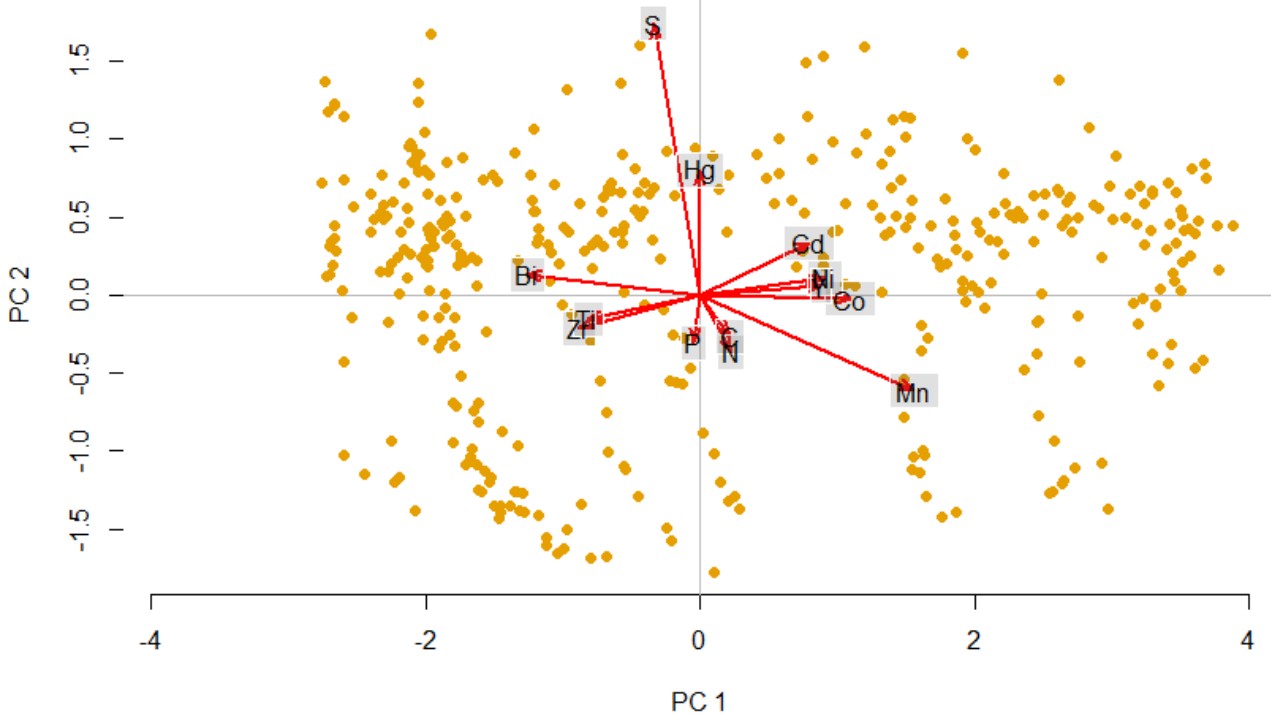

**Figure 3. Biplot for robust compositional PCA of 54 elements in sediments deposited after the year 1960. Thirteen elements with the strongest loadings are shown by arrows.**

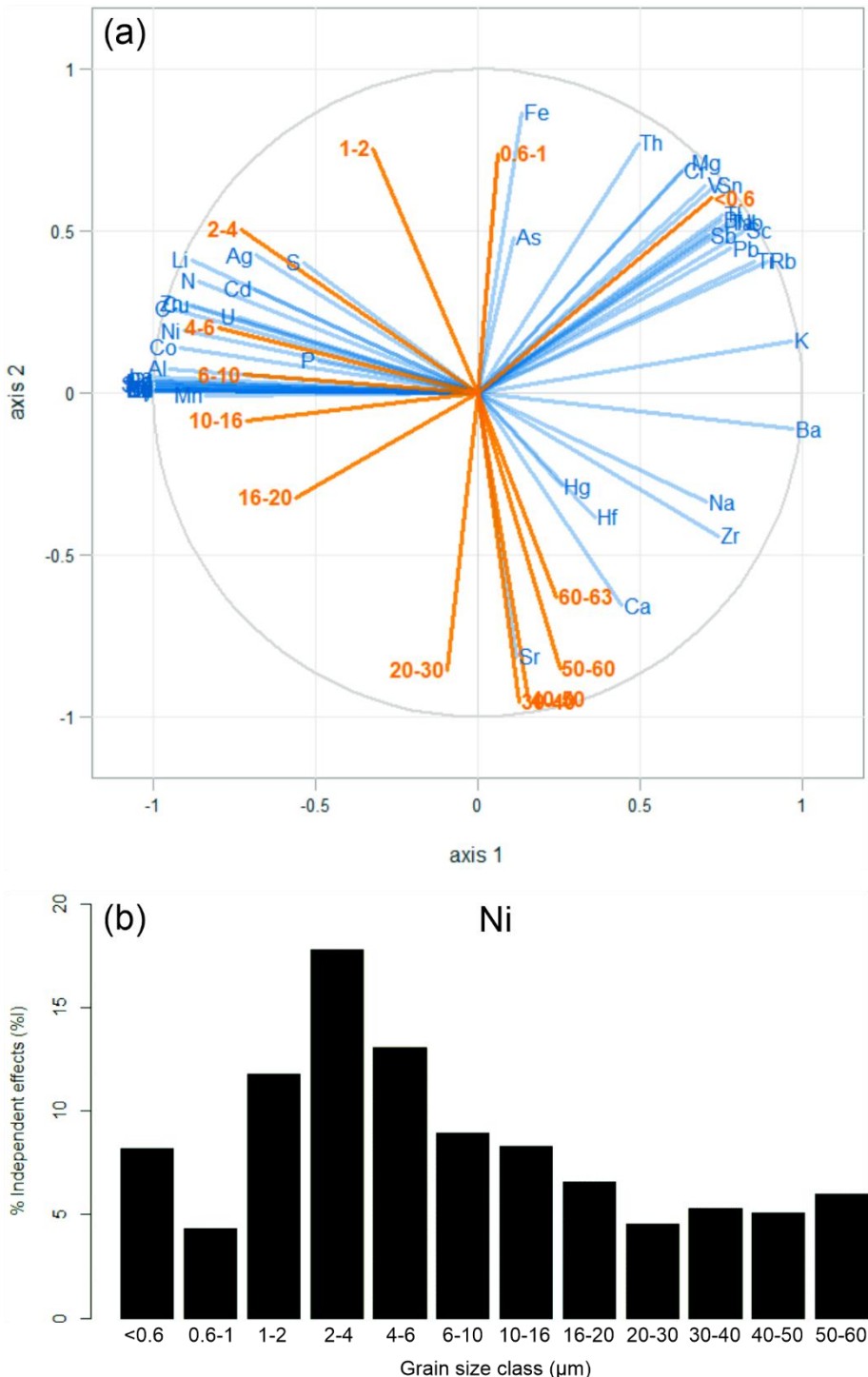

**Figure 4. (a)** Correlation loading plot of the PLSR2 analysis for 54 predictor variables (blue, elements) and 13 response variables (orange, grain size classes in μm). **(b)** Hierarchical partitioning result plot, showing the independent contribution (%) of each grain size class to predicted Ni contents.

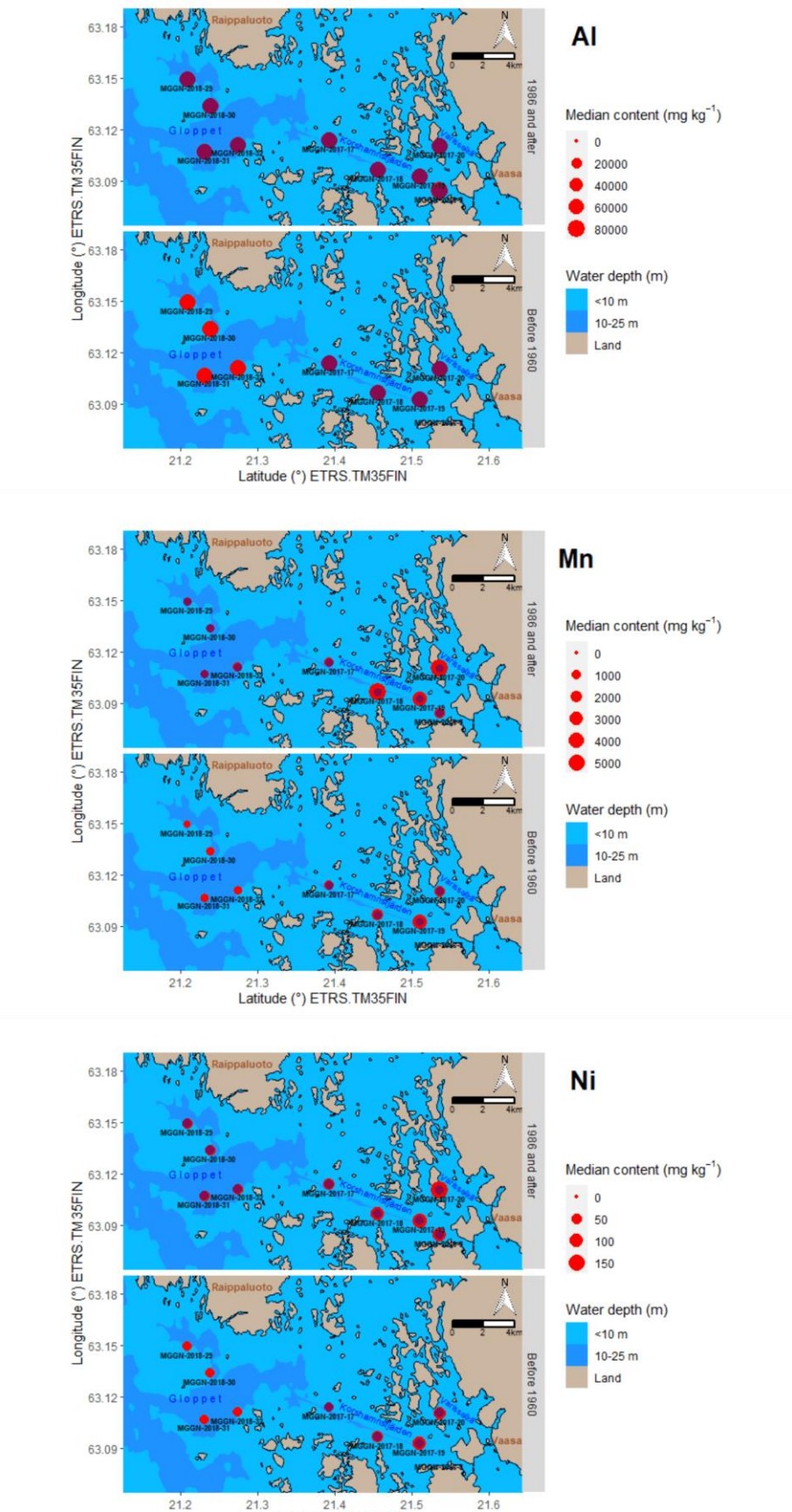


**Figure 5. Map of the sea area off the town of Vaasa with median Al, Mn and Ni contents in the <63 μm grain size fraction of core sections deposited before 1960 (lower panel) and in 1986 and after (upper panel) indicated. Red dots indicate median contents, whereas the contained dark red dots indicate mean median contents in the pre-1960 sections of four cores from the Gloppet area (MGGN-2018-29, MGGN-2018-30, MGGN-2018-31, MGGN-2018-32) in order to highlight the magnitude of enrichment. Note that**
**core MGGN-2016-8 does not contain sediments deposited before 1960. The Laihianjoki and Sulvanjoki rivers are outside the map area, in the southeast corner. Nautical chart: S-57 Finnish Transport Agency 2017.**

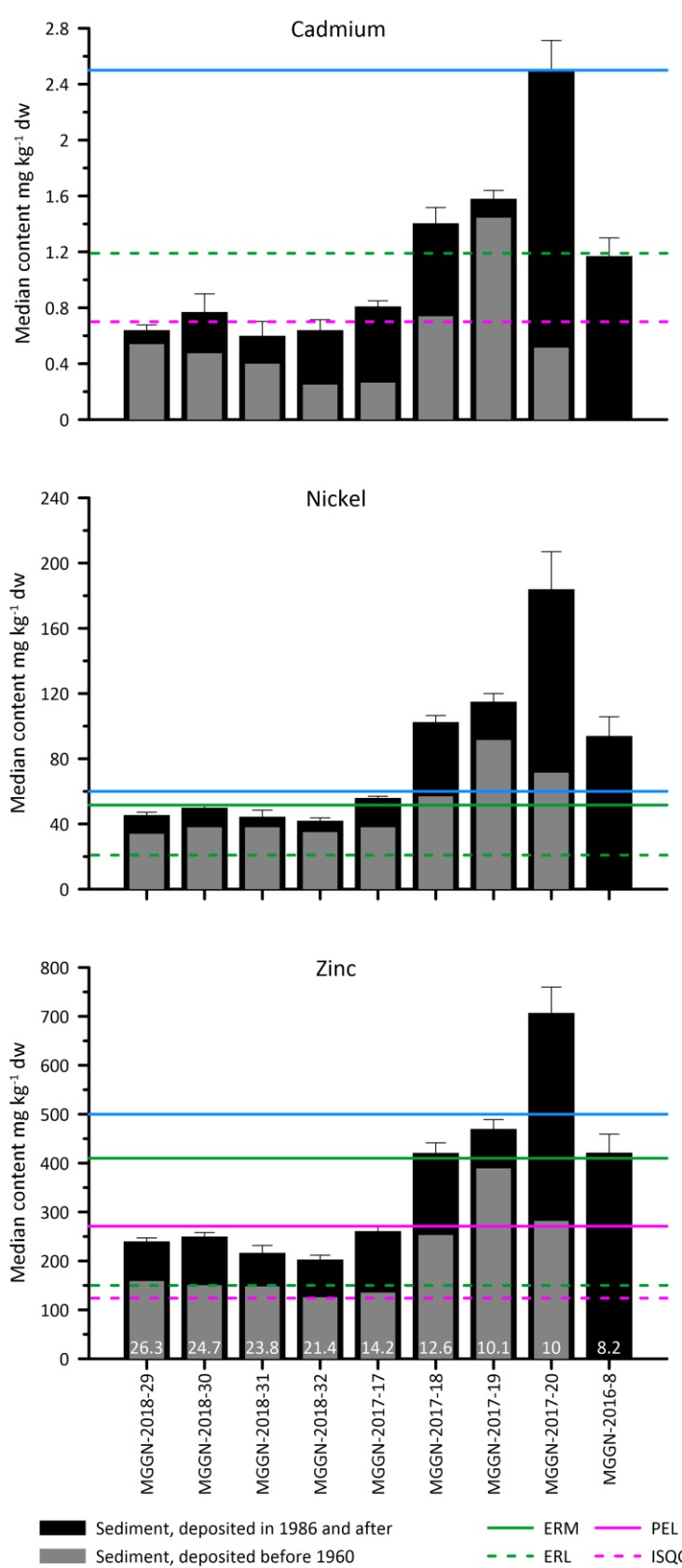

**Figure 6. Median Cd, Ni and Zn contents in sediment cores, arranged according to distance from the Laihianjoki and Sulvanjoki rivers, with sediment quality guidelines indicated. Black bars show median contents in core sections deposited in 1986 and later, whereas the narrower grey bars show contents in core sections deposited before 1960. Whisker lines above the black bars indicate the upper quartile of contents. Note that core MGGN-2016-8 does not contain sediments deposited before 1960. White numbers in the Zn panel denote distance to the closest river mouth in km. Blue line indicates level 2 threshold in Finnish sediment dredging and dumping guidelines (Ympäristöministeriö, 2015). Green solid and dashed lines indicate North American effects range medium (ERM) and effects range low (ERL) thresholds, respectively (Long et al., 1995). Purple solid and dashed lines indicate Canadian probable effect level (PEL) and interim sediment quality guideline (ISQG) thresholds, respectively (Canadian Council of Ministers of the Environment, 2001).**

**Table 1**. Sediment coring locations, water depths, coring dates, core lengths, and distances to the nearest river mouth.

| Sediment core | Latitude ETRS-TM35FIN | Longitude ETRS-TM35FIN | Water depth (m) | Coring date | Core length (cm) | Distance to river (km) | Sedimentation rate (mm yr$^{-1}$) |
|---|---|---|---|---|---|---|---|
| MGGN-2016-8 | 63°05.040 | 21°32.182 | 6 | 4 Aug 2016 | 40 | 8.2 | >13 |
| MGGN-2017-17 | 63°06.828 | 21°23.579 | 10 | 3 July 2017 | 38 | 14.2 | 5.3 |
| MGGN-2017-18 | 63°05.781 | 21°27.339 | 8 | 3 July 2017 | 40 | 12.6 | 6.9 |
| MGGN-2017-19 | 63°05.557 | 21°30.567 | 7 | 3 July 2017 | 40 | 10.1 | 4.0 |
| MGGN-2017-20 | 63°06.640 | 21°32.186 | 5 | 3 July 2017 | 40 | 10.0 | 4.4 |
| MGGN-2018-29 | 63°08.983 | 21°12.564 | 17 | 8 Aug 2018 | 30 | 26.3 | 4.8 |
| MGGN-2018-30 | 63°08.023 | 21°14.327 | 14 | 8 Aug 2018 | 40 | 24.7 | 2.7 |
| MGGN-2018-31 | 63°06.411 | 21°13.894 | 17.5 | 10 Aug 2018 | 40 | 23.8 | 2.3 |
| MGGN-2018-32 | 63°06.675 | 21°16.433 | 15 | 10 Aug 2018 | 42 | 21.4 | 1.7 |

* Sedimentation rate is calculated on the basis of the depth of $^{137}$Cs peak activity due to the 1986 Chernobyl disaster in each core.




| Sediment core | Latitude ETRS-TM35FIN | Longitude ETRS-TM35FIN | Water depth (m) | Coring date | Core length (cm) | Distance to river (km) | Sedimentation rate (mm yr$^{-1}$) |
|---|---|---|---|---|---|---|---|

**Table 2.** Median metal and carbon contents in sediment samples, with interquartile ranges in brackets.

| Core / Study area | n | Al | Cd | Co | Cu | La | Mn | Ni | Zn | C | Reference |
|---|---|---|---|---|---|---|---|---|---|---|---|
| *MGGN-2016-8* | | | | | | | | | | | |
| 1986 and after | 40 | 83650 (72750-87300) | 1.17 (1.09-1.3) | 70.1 (63.2-80.1) | 58.5 (53.7-62.3) | 187.6 (165.7-201.9) | 1605 (1422-1893) | 94.0 (80.0-105.8) | 421.5 (372.5-459.3) | 5.63 (5.16-5.87) | This study |
| *MGGN-2017-17* | | | | | | | | | | | |
| 1986 and after | 17 | 66100 (65400-66900) | 0.81 (0.71-0.85) | 52.8 (45.5-55.8) | 31.0 (30.0-31.0) | 71.7 (70.7-73.6) | 1130 (1080-1640) | 56.0 (54.0-57.0) | 261.0 (257.0-269.0) | 2.74 (2.57-3.05) | This study |
| 1960–1985 | 15 | 66400 (65600-66650) | 0.91 (0.75-1.08) | 40.0 (37.1-43.1) | 29.0 (26.0-31.0) | 61.2 (55.0-63.5) | 954 (921-1030) | 50.0 (45.5-53.0) | 235.0 (194.0-244.5) | 2.18 (2.11-2.22) | |
| Before 1960 | 6 | 67250 (66300-68050) | 0.27 (0.24-0.31) | 23.9 (22.5-25.3) | 23.0 (22.3-23.0) | 47.3 (46.5-49.6) | 948 (908-1009) | 38.0 (35.0-38.8) | 135.5 (135.0-139.0) | 2.04 (1.99-2.06) | |
| *MGGN-2017-18* | | | | | | | | | | | |
| 1986 and after | 22 | 69700 (67950-71900) | 1.41 (1.23-1.52) | 97.9 (83.5-100.0) | 40.0 (39.0-41.0) | 120.0 (111.5-124.8) | 5505 (3823-7273) | 102.5 (94.3-106.5) | 421.0 (396.0-441.5) | 4.06 (3.50-4.15) | This study |
| 1960–1985 | 17 | 70700 (69200-71900) | 1.48 (1.18-1.58) | 63.4 (59.6-66.8) | 40.0 (33.0-43.0) | 98.2 (80.6-103.0) | 2350 (1810-2870) | 84.0 (77.0-88.0) | 369.0 (318.0-384.0) | 3.29 (2.89-3.36) | |
| Before 1960 | 1 | 66900 | 0.74 | 43.9 | 29.0 | 66.1 | 1700 | 57.0 | 253.0 | 2.63 | |
| *MGGN-2017-19* | | | | | | | | | | | |
| 1986 and after | 13 | 75200 (74100-76100) | 1.58 (1.48-1.64) | 95.3 (86.9-105.0) | 44.0 (43.0-46.0) | 151.0 (139.0-156.0) | 3510 (3390-5990) | 115.0 (102.0-120.0) | 470.0 (441.0-489.0) | 4.53 (4.10-4.89) | This study |
| 1960–1985 | 11 | 75000 (74200-75550) | 1.57 (1.34-1.60) | 66.0 (64.7-68.2) | 42.0 (41.5-44.0) | 126.0 (121.5-127.0) | 3010 (2050-4405) | 86.0 (83.5-88.0) | 377.0 (362.5-381.5) | 3.76 (3.75-3.92) | |
| Before 1960 | 16 | 74400 (73350-74900) | 1.45 (1.41-1.64) | 70.7 (63.5-74.6) | 42.5 (40.0-44.3) | 107.5 (100.8-117.5) | 3305 (2333-5240) | 91.5 (84.0-95.3) | 389.5 (371.5-411.0) | 3.64 (3.54-3.77) | |
| *MGGN-2017-20* | | | | | | | | | | | |
| 1986 and after | 14 | 73350 (72750-74700) | 2.51 (2.39-2.71) | 164.0 (153.7-180.5) | 55.0 (57.0-62.0) | 209.0 (192.5-219.5) | 5600 (4965-7280) | 184.0 (175.3-207.0) | 707.0 (687.8-760.0) | 5.92 (5.79-6.00) | This study |
| 1960–1985 | 12 | 70950 (70500-75425) | 2.17 (1.57-2.57) | 121.5 (92.93-166.8) | 45.5 (32.5-56.8) | 128.0 (102.3-173.5) | 2660 (1948-5990) | 146.0 (111.0-183.3) | 532.0 (426.5-662.5) | 3.32 (2.88-4.38) | |
| Before 1960 | 14 | 69050 (68400-69875) | 0.52 (0.45-0.71) | 56.60 (51.23-62.78) | 26.0 (25.0-27.0) | 74.3 (71.3-76.8) | 1630 (1478-1823) | 71.5 (69.0-75.8) | 282.0 (258.5-289.3) | 2.41 (2.35-2.45) | |
| *MGGN-2018-29* | | | | | | | | | | | |
| 1986 and after | 16 | 70250 (68450-70625) | 0.64 (0.58-0.68) | 29.1 (26.4-31.7) | 36.0 (33.8-37.3) | 65.1 (63.6-66.3) | 543 (524-592) | 45.5 (42.5-47.3) | 240.0 (216.3-247.0) | 3.27 (3.21-3.40) | This study |
| 1960–1985 | 13 | 68000 (65900-69600) | 0.86 (0.71-1.05) | 24.3 (20.4-26.5) | 37.0 (36.0-37.0) | 56.4 (54.9-59.3) | 584 (564-607) | 41.0 (40.0-41.0) | 243.0 (214.0-250.0) | 3.03 (2.76-3.10) | |
| Before 1960 | 1 | 71200 | 0.54 | 19.60 | 27.0 | 49.2 | 542 | 34.0 | 159.0 | 2.37 | |
| *MGGN-2018-30* | | | | | | | | | | | |
| 1986 and after | 9 | 69500 (68500-70600) | 0.77 (0.60-0.90) | 40.2 (36.1-43.5) | 33.0 (31.0-35.0) | 68.2 (66.3-68.8) | 718 (683-1250) | 50.0 (48.0-52.0) | 250.0 (228.0-258.0) | 3.07 (3.04-3.39) | This study |
| 1960–1985 | 7 | 69600 (69400-70500) | 1.00 (0.80-1.34) | 32.0 (31.2-37.1) | 34.0 (30.5-34.5) | 58.3 (55.3-62.8) | 621 (620-641) | 46.0 (43.5-48.0) | 225.0 (213.5-266.5) | 2.57 (2.54-2.71) | |
| Before 1960 | 24 | 68950 (68150-70000) | 0.48 (0.35-0.54) | 21.7 (21.1-23.7) | 26.0 (25.0-27.0) | 49.0 (48.6-49.8) | 726 (700-757) | 38.0 (37.0-39.0) | 150.5 (140.0-163.8) | 2.36 (2.27-2.37) | |
| *MGGN-2018-31* | | | | | | | | | | | |
| 1986 and after | 8 | 69800 (67600-71525) | 0.60 (0.52-0.70) | 28.9 (27.9-31.7) | 32.0 (29.5-34.0) | 64.5 (61.6-65.4) | 695 (625-1205) | 44.5 (42.0-48.5) | 216.5 (199.8-231.8) | 3.33 (3.15-3.41) | This study |
| 1960–1985 | 7 | 70800 (70550-71600) | 0.85 (0.65-0.98) | 29.7 (28. 3-33.3) | 34.0 (32.5-37.0) | 55.5 (55.2-58.2) | 551 (547-573) | 45.0 (44.0-45.5) | 224.0 (208.5-248.5) | 2.63 (2.61-2.76) | |
| Before 1960 | 25 | 70600 (69600-71600) | 0.40 (0.27-0.55) | 21.6 (19.5-23.1) | 27.0 (27.0-29.0) | 49.7 (48.9-50.9) | 628 (614-642) | 38.0 (36.0-39.0) | 148.0 (135.0-163.0) | 2.36 (2.27-2.40) | |
| *MGGN-2018-32* | | | | | | | | | | | |
| 1986 and after | 6 | 67350 (66025-68150) | 0.64 (0.54-0.72) | 32.7 (30.2-35.7) | 27.0 (26.3-29.3) | 59.5 (59.3-60.2) | 909 (605-2085) | 42.0 (40.3-43.8) | 203.0 (193.5-211.8) | 2.70 (2.58-2.99) | This study |
| 1960–1985 | 6 | 69400 (68650-70825) | 0.56 (0.54-0.75) | 29.9 (25.2-31.9) | 26.5 (26.0-28.5) | 51.3 (49.6-54.2) | 557 (535-567) | 40.0 (37.0-43.8) | 180.5 (159.8-199.8) | 2.28 (2.21-2.40) | |
| Before 1960 | 30 | 69850 (69025-70925) | 0.25 (0.21-0.32) | 18.7 (17.6-19.6) | 24.0 (24.0-25.0) | 48.7 (47.6-49.4) | 633 (607-652) | 35.0 (34.0-37.0) | 126.0 (120.5-133.3) | 2.07 (1.92-2.14) | |
| *Total* | | | | | | | | | | | |
| 1986 and after | 145 | 71100 (67900-75200) | 1.10 (0.75-1.40) | 68.5 (42.6-93.0) | 41.0 (33.0-55.0) | 125.0 (67.8-175.8) | 1800 (1120-4090) | 88.0 (51.0-107.0) | 389.0 (254.0-454.0) | 4.10 (3.26-5.39) | This study |

| | | n | | | | | | | | | | Reference |
|---|---|---|---|---|---|---|---|---|---|---|---|---|
| | 1960–1985 | 88 | 70250 (67925-71950) | 1.08 (0.83-1.52) | 43.1 (30.8-66.2) | 35.5 (30.0-41.3) | 64.4 (56.3-107.0) | 1030 (595-2355) | 54.5 (43.7-86.0) | 260.0 (222.5-374.8) | 2.79 | |
| 905 | Before 1960 | 117 | 69900 (68600-71400) | 0.44 (0.27-0.62) | 21.7 (19.5-41.0) | 26.0 (24.0-28.0) | 49.7 (48.6-68.5) | 669 (631-1270) | 38.0 (36.0-58.0) | 148.0 (131.0-244.0) | 2.34 | |
| | *Grand total* | 350 | 70300 (68300-72700) | 0.86 (0.53-1.32) | 45.3 (24.3-72.8) | 33.0 (27.0-43.0) | 68.4 (51.8-124.8) | 1195 (644-2490) | 55.0 (40.0-91.0) | 258.0 (176.2-393.5) | 3.04 | This study |
| 910 | *Laihianjoki Estuary* (2015) Surface sediment, microwave $HNO_3$ digestion | 1 | 59900 | 0.92 | 74.3 | 63.1 | n.d. | 788 | 130.5 | 461 | n.d. | Wallin et al. |
| 915 | *Vöyrinjoki Estuary* 2008b Site B, 1960 and after Four acid digestion | 13 | 86400 (82900-87600) | 1.54 (1.32-1.76) | 112.1 (106.6-128.6) | 59.9 (58.6-64.0) | 187 (186-198) | 9013 (6854-10923) | 83.9 (80.1-98.5) | 529.1 (493.6-592.0) | n.d. | Nordmyr et al. |
| | *Bothnian Bay* Surface sediment, hydrofluoric acid digestion, mean content | 5 | n.d. | 0.8 ±0.3 | n.d. | 46 ±15 | n.d. | 8500 ±5300 | n.d. | 216 ±92 | 4.0 ±1.2 | Leivuori and Niemistö, 1995 |
| | *Bothnian Sea* Surface sediment, hydrofluoric acid digestion, mean content | 5 | n.d. | 0.4 ±0.2 | n.d. | 30 ±11 | n.d. | 3000 ±1600 | n.d. | 173 ±56 | 2.3 ±1.0 | Leivuori and Niemistö, 1995 |

920   Metal contents are in mg kg$^{-1}$ dry weight. Carbon content is in % dry weight. N.d. denotes not determined.