# Peer review of "Enrichment of trace metals from acid sulphate soils in sediments of the Kvarken Archipelago, eastern Gulf of Bothnia, Baltic Sea"

_Biogeosciences, 2020_

## Short Comment (SC1) · 17 Sep 2020

September 17, 2020

Review of manuscript bg_2020_231

Manuscript title: Enrichment of trace metals from acid sulphate soils in sediments of the Kvarken Archipelago, eastern Gulf of Bothnia, Baltic Sea

Authors: Virtasalo, J. J., Österholm, P., Kotilainen, A. T., Åström„ M. E.

RECOMMENDATION: Accept for publication with minor revision

[Figure]

**SYNOPSIS AND GENERAL COMMENTS**

Synopsis The manuscript investigates the spatial and temporal distribution of metals in nine sediment cores from the Kvarken Archipelago, which is affected by drainage of acid sulphate soils (ASS) in western Finland. The manuscript assesses the trace metal distribution in Gulf of Bothnia sediments with increasing distance from river mouths. The focus is on sediments deposited before and after the intensive artificial drainage of ASS that began in the 1960s. Focus is also on metals that are leached from ASS (Al, Cd, Co, Cu, La, Mn, Ni, and Zn). There is a lack of investigations regarding metal transport from ASS and through the estuarine zone of rivers in the Gulf of Bothnia, and the topic is well in line with the Aims and Scope of Biogeosciences.

Novelty and scientific merit / significance The study does not present any new methods or novel approaches in the interpretation of data. Although the study primarily may be important from a local point of view, results on the transport of metals from ASS through the coastal zone apparently have not been assessed previously. The study thus has a scientific merit, and is of general interest to researchers studying metal release from ASS.

Methodology and quality of work Sampling and measurements Sampling and analytical methods used in the study are suitable and appropriate.

For metals, the analytical accuracy is reported relative to CRMs and in-house standards. The authors should also report the analytical precision for the metals. This could be critical for interpretation of some of the metal profiles shown in Fig. 2.

Aim, interpretations and conclusions Define the aim/objective more clearly. Was the aim to assess the metal distribution in sediments with increasing distance from rivers? This is what the interpretations and conclusions are linked to.

In general the interpretations are supported by the results. However, the manuscript should briefly state the arguments supporting that the metal increase from the 1960s is

really caused by leaching from ASS, and not from industrial sources. These arguments are probably reported in some of the cited references (e.g. Nordmyr et al), but could be briefly mentioned in the manuscript to strengthen the conclusions.

Readability and language The manuscript is well written and easy to read, and the abstract accurately reflects the content of the manuscript.

SPECIFIC COMMENTS

Line Comment 57 Should read: The metal distribution ... 134 I cannot se that Hg is discussed in the manuscript, so it can be removed from the methods section. 247– 249 Lines can be deleted. This is already mentioned in the Introduction. 345 The fact that Cd, Co, Cu, Ni, Zn correlate with the same grain sizes as C and N does not automatically imply a causal correlation. This may well be the case, but I suggest using the term "suggest" instead of "indicate".

Supplementary file Title in Supplement should agree with title of the manuscript ("trace" and "Baltic Sea" are missing)

Figures Figure 5: Show "contained dark red dots" in another colour. They are now very difficult to see. This also applies to the figures (maps) in the Supplement.

Tables Table 1: Report the sedimentation rate (cm/year) for each core

References Line 323: Should Cook et al. be 1997 or 2000?

---

## Referee Comment (RC1) · Thomas Job (Referee) · 21 Sep 2020

General comments:

This work contributes to important scientific questions regarding the transport and fate of metals mobilised from oxidised acid sulfate soils, which fall within the scope of Biogeosciences.

While not a novel concept, per se, the results and conclusions of this paper are important contributions to understanding the source-sink transport of contaminants in estuarine-marine systems which, to date, is poorly constrained due to complex biogeo-

chemistry and the interconnectivity of sediment transport processes, geomorphology, hydrology, and climate.

The sampling design, analytical methods, and statistical analyses are robust and appropriate.

Specific comments:

Research aims should be more explicitly presented in the final paragraph of the introduction.

The role of Fe-oxyhydroxides in metal sequestration is mentioned in line 48, however the discussion of Fe is absent from this paper. Data from Australia point to Fe species as a major product of acidic drainage, and a major sink for mobilised metals (see Bush et al., 2004; Mosely et al., 2018; Job et al., 2020). I recognise that some of the papers you reference do not observe high levels of dissolved Fe in stream waters affected by acidic drainage, but Fe could be transported in solid phases (near-source precipitation), and subsequently still be accumulating at elevated levels in this terminal system. If you have data to the contrary, stating this would be a helpful insight to international audiences.

Bush, R. T., Fyfe, D. and Sullivan, L. A. (2004) 'Occurrence and abundance of monosulfidic black ooze in coastal acid sulfate soil landscapes', Australian Journal of Soil Research, 42(5–6), pp. 609–616. doi: 10.1071/sr03077.

Mosley, L. M. et al. (2018) 'Fate and dynamics of metal precipitates arising from acid drainage discharges to a river system', Chemosphere, 212, pp. 811–820. doi: https://doi.org/10.1016/j.chemosphere.2018.08.146.

Job, T., Penny, D. and Morgan, B. (2020) 'Geochemical signatures of acidic drainage recorded in estuarine sediments after an extreme drought', Science of The Total Environment, 749, p. 141435. doi: https://doi.org/10.1016/j.scitotenv.2020.141435.

Regarding the correlation of trace metals Cd, Co, Cu, Ni, and Zn to the grain size

fraction (2–6 $\mu$m), and to C&N, which you conclude to represent metal-organic matter aggregates; can you please clarify how this relationship can be established when the grain-size analysis method digests organic matter? Also, why might the relationship between C&N with trace metals be less evident in the robust PCA analysis?

In lines 265-285, when discussing the Manganese data, the impact of redox transformations on the down-core element profile is mentioned. Do you have any data on redox conditions in the cores? The colour change in the core photos is notable. An acknowledgement of how redox transformations may be impacting the geochemical profiles of the other elements would be of benefit. The redox transition appears to be at notably shallower core depths than the 1986 temporal marker.

Redox states are also of significance to the Risk Assessment section where redox conditions have implications for bioavailability (the presence of AVS for example). I suggest it is acknowledged in this section that further data is required to determine the bioavailability of these metals (dilute acid-extractable, for example). The magnitude of enrichment is certainly sufficient to flag potential ecotoxic risk however.

A brief description of changes in lithology observable in the cores would be of benefit in the primary manuscript (I recognise that core photos are included in the supplementary material – this could be referenced in-text), at the very least to exclude sedimentological changes as a primary driver of geochemical variability.

It is identified that the 2–6 $\mu$m grain-size fraction is positively associated with trace metal loading – how variable is this grain-size fraction down-core? Regarding sediment transport, OM accumulating with 2–6 $\mu$m siliciclastics or carbonates would presumedly be coarser in grain-size due to density differences. A down-core log of any grain-size variability may similarly help interpret controls on down-core geochemical variability.

An assessment of experimental precision should be included. Were replicates analysed?

[Figure]

Lines 43-44: Provide a reason why climate change might increase acid release from AS Soils.

Lines 59-67: I think the importance of understanding contaminant dynamics should be more explicit / clear – it is a valuable contribution of this research.

Line 108: . . . cool and dark conditions - include the approximate temperature of storage, as well as time until laboratory analysis. Where/how were the samples stored in the shore-based laboratory?

Line 171: the grain-size distributions are described as poorly sorted but from a narrow grain-size range. This sounds contradictory (poor sorting implies wide grain-size range) however I think the intention is that the median grain-sizes exhibit low variability. Please clarify, and consider including the range of sorting measurements.

Line 334: precipitation does not only occur 'out to sea' - explicitly identify neutralization, which can coincide with reaching seawaters.

Technical Comments:

The term contents is regularly used, when perhaps concentration would be more explicit and clear. Consider amending.

Line 12: which is the recipient system of the Laihianjoki and Sulvanjoki . . .

Line 14: . . . landscape. Metal deposition has remained at high levels since . . .

Line 26: . . . low pH conditions in pore- and surface-waters . . .

Lines 28-30: Due to the uncertainty, perhaps simplify these two sentences into one.

Line 54: deteriorative*

Line 63: in the long term

Line 111: Define 'fresh'.

Line 118: . . . classified as those . . .

Line 128: . . . dissolved in 1 M HNO3. . .

Line 130: Instead identify which element was analysed by which technique inside brackets.

Line 168: . . . in the R software environment.

Line 195: Mn contents are low except for a strong increase

Line 241: the pattern of decreasing metal contents with increasing distance

Line 242: The median metal contents . . .

Line 251: Reword this sentence as it implies uniformity both laterally and vertically, which you go on to clarify is not the case.

Lines 287-300 – here, and some other sections of the discussion, are moderately convoluted with data, impacting readability. Some of this could be shifted into the results section, or just simplified.

Line 307: begin to decrease

Line 312: as a result of, for example, . . . (the abbreviation here breaks readability)

Line 312-313: rather than a decrease

Lines 316-323 – I do not think this paragraph is necessary. It is mostly already explained earlier in the manuscript.

Line 401: ecotoxicological*

---

## Author Comment (AC1) · 2 Oct 2020

Response to Short Comment by Anders Widerlund

We thank Anders Widerlund for insightful and constructive comments that help improve the manuscript. His main concerns will be addressed as follows.

Analytical precision: We have done replicate analyses of several samples, and will add a statement about experimental precision in the revised manuscript.

We will elaborate the significance of leaching from acid sulphate soils since the 1960s as the primary source of the studied metals, instead of e.g. industrial sources. This is

mentioned in the final sentence of Introduction, but it obviously needs to be elaborated more in the revised manuscript.

We will elaborate the inferred relationship between trace metals from acid sulphate soils, 2-6 $\mu$m grain size fraction, and C&N in the revised manuscript. The statistical relationship is shown by two independent methods. However, we agree with Widerlund that the statistical relationship rather "suggests" than "indicates" a causal correlation. At least until additional data is available.

We will correct the title of the Supplementary file as correctly pointed out by Widerlund.

We will change the colour of "contained dark red dots" in Figure 5 and in the Supplement, as Widerlund suggests.

We will add sedimentation rate for each core in Table 1, as Widerlund suggests.

The rest of Widerlund's comments are justified but comparably minor, and can be fully addressed in the revised manuscript.

Kind regards on behalf of all co-authors,

Joonas Virtasalo, Geological Survey of Finland

---

## Author Comment (AC2) · 2 Oct 2020

Response to Referee #1 Thomas Job

We thank referee Thomas Job for insightful and constructive comments that help improve the manuscript.

Referee has several Specific Comments, which are well justified, and which we will address as follows.

Fe-oxyhydroxides were not considered as metal carriers in the paper, because Fe typically has low relative mobility and transport in Finnish (boreal) soils (c.f., Nordmyr et

al. 2008, Marine Environmental Research). This is probably due to precipitation of Fe-oxyhydroxides in soil cracks after oxidation of sulfides, which inhibits their release from the soils. Moreover, Fe that eventually makes it to the streams, is to a large part deposited as Fe-oxyhydroxides close to river mouths (Nystrand et al. 2016, Applied Geochemistry), making it less significant offshore metal carrier. Referee is correct that Fe-oxyhydroxides deserve more attention. We will elaborate Fe behavior in the revised manuscript, and add a map of Fe contents at our coring sites in order to satisfy the international audience.

We will elaborate the inferred relationship between trace metals from acid sulphate soils, 2-6 $\mu$m grain size fraction, and C&N in the revised manuscript. The 2-6 $\mu$m lithic particles are interpreted to be constituents of the original organic-rich aggregates that were broken-up in sediments. The statistical relationship, shown by two independent methods, suggest that these organic rich aggregates were important seaward carriers of the trace metals. The weaker but existing statistical relationship between the trace metals and C&N in the PCA probably reflects marine (authigenic) source of organic material, in addition to the organic-rich aggregates that form at the river mouth.

Mn enrichment in the core tops further out at sea is a consequence of the high redox-driven mobility of Mn in the sediments. We will explain in the revised manuscript that comparable enrichment patterns were not observed for other trace metals in the studied cores. These sediments have high organic contents, and the redox transition typically is very steep, which means that newly deposited sediment is soon buried into the reducing part of the sediment column. We will explain this in the revised manuscript, but do wish to note that redox processes are not the main topic of this study, and they also are not very relevant for our interpretations and conclusions (except for Mn).

Referee is correct that bioavailability of metals is influenced by sediment redox state. We will add a statement that further data is required to determine the bioavailability of the enriched metals in the revised manuscript, as Referee suggests.

We will add a brief description of core lithologies in the primary manuscript in the revised version. Referee is correct that this is relevant information was missing.

We will add down-core logs of grain-size variability, and describe grain-size trends in the revised manuscript, as Referee suggests.

We have replicate analyses of several samples. We will add a statement about experimental precision in the revised manuscript. Referee is correct that this is relevant information was missing.

The rest of Referee's Specific Comments are justified but comparably minor, and can be fully addressed in the revised manuscript.

Referee's Technical Comments can be followed as such.

Kind regards on behalf of all co-authors,

Joonas Virtasalo, Geological Survey of Finland

---

## Referee Comment (RC2) · Anonymous Referee #2 · 8 Oct 2020

This is a carefully prepared manuscript on trace metals from acid sulphate soils in sediments from the Baltic Sea. I have only a few minor comments, see below.

Line 11. I would suggest to focus on the sites not the cores: "in sediments at 9 sites in the Kvarken Archipelago"

Line 14. Suggested change: "a high level"

Line 18. Suggested change: "in the same sediment"

Line 29. Is it relevant to mention that the soils are currently being mapped? I would propose to combine the two last sentences of this paragraph: "In Europe, the largest

occurrences of AS soils are probably found in Finland where current estimates point to AS soils occupying an area in the order of 1 million ha (Anton Boman, personal communication)."

Line 33. This sounds as if the brackish phase started in the Gulf of Bothnia, whereas the salt water entered from the south. This could be solved by saying: ". . .phase, which, in the Gulf of Bothnia, began ca. 7000 years ago" or something similar.

Line 36: suggested change: "a significant lowering"

Line 39: suggested change: "extremely acidic"

Line 42: suggested change: "for biodiversity"

Line 58. suggested change: "the distribution pattern of the metals"

Line 62. What does "expanding dredging" mean? Can you rephrase?

Line 69. How is the distance defined here? The distance to the river mouth?

Line 71. "from the Laihianjoki . . ." Why is river in capitals here? Is it part of the name?

Line 93. Salinity has no units (it is defined relative to the conductivity of KCl), so PSU is not necessary here.

Line 95. suggested change: "a thermocline"

Line 100. suggested change: "the summers of 2016-2018"

Line 108. suggested change: "in the cold and dark"

Line 126. Why was a sieve fraction used for this analysis? Since this treatment leads to higher element concentrations, comparison to other studies may not be not directly possible, unless the fraction >63 um was small. That only the fraction <63 um was analysed needs to be mentioned in the captions of figures 2 and 5.

Line 128. Suggested change "in HNO3"

Line 170. I suggest to include the results of the grain size analysis in a supplement.

Line 191: Suggested change "similar to that of other metals"

Line 195: Suggesetd change: "except for a strong"

Line 344-345. Here the authors write: "Also the nutrients C and N have strong positive correlations with the same grain-size range, which indicates that the metals are associated with organic particles". This is too strongly formulated. Correlation does not imply causation. I would suggest to replace "indicates" by "suggests"

Line 418-420. See the previous point. "The strong association of the metals and nutrients to sediment grains of the same size range (2–6 $\mu$m) indicates that the transformation of dissolved organic matter and metals to metal-organic aggregates at the river mouths is the key mechanism of seaward trace metal transport." Indicates should be replaced by "suggests"

---

## Author Comment (AC3) · 9 Oct 2020

Response to Referee #2

We thank Referee #2 for insightful and constructive comments that help improve the manuscript.

Referee's comments are justified and comparably minor. A few of comments require more consideration than changes to wording. These will be considered as follows.

The use of the sieve fraction <63 $\mu$m for multielement analyses is a common practice in geochemical analysis because cations generally are adsorbed onto fine particles.

The studied sediments are fine grained, with the median grain sizes ranging between 2 and 3 $\mu$m. Only one sample had more than 1 % grains larger than 63 $\mu$m (6.7 %). Because of the insignificant contribution of grains larger than 63 $\mu$m in our samples, the analysis of sieved fraction is not considered to have significantly impacted our results or conclusions, as also suggested by Referee.

Multielement and grain size analysis results will be published in the PANGAEA online data archive after this manuscript has been accepted for publication.

We will elaborate the inferred relationship between trace metals from acid sulphate soils, 2-6 $\mu$m grain size fraction, and C&N in the revised manuscript. The statistical relationship is shown by two independent methods. However, we agree with Referee that the statistical relationship rather "suggests" than "indicates" a causal correlation.

The rest of Referee's comments can be fully addressed in the revised manuscript.

Kind regards on behalf of all co-authors,

Joonas Virtasalo, Geological Survey of Finland

---

## Author Response (AR1)

Dear Editor,

Thank you for considering our manuscript "Enrichment of trace metals from acid sulphate soils in sediments of the Kvarken Archipelago, eastern Gulf of Bothnia, Baltic Sea" for publication in *Biogeosciences*.

We have now revised the manuscript according to the comments by Referee #1 Thomas Job, Referee #2 anonymous, and Dr. Anders Widerlund. In general, we found the comments justified, insightful and helpful with respect to improving the manuscript.

Below we answer all review comments point by point. Reviewer comments are in **black**, and our responses in blue.

A marked-up version of the revised manuscript is appended at the end of this document.

Line numbers given in brackets **[]** in our responses to review comments refer to the line numbers in the marked-up manuscript.
* * *
**Referee #1 Thomas Job:**

**General comments:**

**This work contributes to important scientific questions regarding the transport and fate of metals mobilised from oxidised acid sulfate soils, which fall within the scope of Biogeosciences.**

**While not a novel concept, per se, the results and conclusions of this paper are important contributions to understanding the source-sink transport of contaminants in estuarine-marine systems which, to date, is poorly constrained due to complex biogeo-chemistry and the interconnectivity of sediment transport processes, geomorphology, hydrology, and climate.**

**The sampling design, analytical methods, and statistical analyses are robust and appropriate.**

We thank Referee for the encouraging words, and the insightful and constructive review of our manuscript.

**Specific comments:**

**Research aims should be more explicitly presented in the final paragraph of the introduction.**

We have now clarified the statement of research aims in that paragraph. **[Line 74]**

**The role of Fe-oxyhydroxides in metal sequestration is mentioned in line 48, however the discussion of Fe is absent from this paper. Data from Australia point to Fe species as a major product of acidic drainage, and a major sink for mobilised metals (see Bush et al., 2004; Mosely et al., 2018; Job et al., 2020). I recognise that some of the papers you reference do not observe high levels of dissolved Fe in stream waters affected by acidic drainage, but Fe could be transported in solid phases (near-source precipitation), and subsequently still be accumulating at elevated levels in this terminal system. If you have data to the contrary, stating this would be a helpful insight to international audiences.**

Bush, R. T., Fyfe, D. and Sullivan, L. A. (2004) 'Occurrence and abundance of monosulfidic black ooze in coastal acid sulfate soil landscapes', Australian Journal of Soil Research, 42(5–6), pp. 609–616. doi: 10.1071/sr03077.

Mosley, L. M. et al. (2018) 'Fate and dynamics of metal precipitates arising from acid drainage discharges to a river system', Chemosphere, 212, pp. 811–820. doi: https://doi.org/10.1016/j.chemosphere.2018.08.146.

Job, T., Penny, D. and Morgan, B. (2020) 'Geochemical signatures of acidic drainage recorded in estuarine sediments after an extreme drought', Science of The Total Environment, 749, p. 141435. doi: https://doi.org/10.1016/j.scitotenv.2020.141435.

That is a good suggestion. We added a paragraph in the Discussion section about the (low) transport of Fe to the sea area, and a growing symbol map of Fe contents in the Supplementary File. **[Lines 343-357]**

**Regarding the correlation of trace metals Cd, Co, Cu, Ni, and Zn to the grain size fraction (2–6 μm), and to C&N, which you conclude to represent metal-organic matter aggregates; can you please clarify how this relationship can be established when the grain-size analysis method digests organic matter? Also, why might the relationship between C&N with trace metals be less evident in the robust PCA analysis?**

We added a sentence highlighting that the 2-6 μm sized particles are likely mineral grains that are commonly found as constituents of organic aggregates. **[Lines 407-411]**

The weaker but existing statistical relationship between the trace metals and C&N in the PCA probably reflects marine (authigenic) source of organic material (phytoplankton), in addition to the organic aggregates that form at the river mouth. Although this is a likely explanation that also conforms with our conclusions, we find it a bit speculative and therefore have not included it in the manuscript.

**In lines 265-285, when discussing the Manganese data, the impact of redox transformations on the down-core element profile is mentioned. Do you have any data on redox conditions in the cores? The colour change in the core photos is notable. An acknowledgement of how redox transformations may be impacting the geochemical profiles of the other elements would be of benefit. The redox transition appears to be at notably shallower core depths than the 1986 temporal marker.**

Cores from the outermost sites show a strong increase of Mn at the core tops, which is attributed to the reduction of Mn oxides under reducing conditions in the sediments, and upward diffusion and oxidative precipitation of Mn as oxyhydroxides in the sediment surface layer. Similar enrichment in the surface layer is not observed for other metals, which is quite as expected because Mn is by far the most redox sensitive of the studied metals from AS soils.

We added a sentence in the manuscript, stating that Mn is the only metal from AS soils that shows such redox-driven migration in the studied cores. **[Lines 310-311]**

**Redox states are also of significance to the Risk Assessment section where redox conditions have implications for bioavailability (the presence of AVS for example). I suggest it is acknowledged in this section that further data is required to determine the bioavailability of these metals (dilute acid-extractable, for example). The magnitude of enrichment is certainly sufficient to flag potential ecotoxic risk however.**

We added a sentence stating that our assessment would benefit from determining the speciation of metals in sediments. Furthermore, instead of individual metals, the combined toxic effects of several metals and environmental factors should be considered. **[Lines 457-460]**

**A brief description of changes in lithology observable in the cores would be of benefit in the primary manuscript (I recognise that core photos are included in the supplementary material – this could be referenced in-text), at the very least to exclude sedimentological changes as a primary driver of geochemical variability.**

Referee is correct that this relevant information was missing in the manuscript. We added a paragraph about the core lithology under the new section 4.1 in the primary manuscript, with a reference to Supplementary File where pictures of the cores are shown. **[Lines 184-190]**

**It is identified that the 2–6 µm grain-size fraction is positively associated with trace metal loading – how variable is this grain-size fraction down-core? Regarding sediment transport, OM accumulating with 2–6 µm siliciclastics or carbonates would presumedly be coarser in grain-size due to density differences. A down-core log of any grain-size variability may similarly help interpret controls on down-core geochemical variability.**

We added down-core logs of median grain size and the share of the 2-6 µm grain size class for each core where available in the Supplementary File. The median grain sizes range between 2 and 3 µm, and the share of the 2-6 µm grains typically is 20-30%. There is thus less variability in the sediment grain size, which strongly indicates that the vertical metal enrichment patterns in the cores as described and discussed in the manuscript are not controlled by autochthonous processes in the sedimentary environment, but by external metal loading from AS soils. We added a statement about this in the manuscript. **[Lines 413-415]**

**An assessment of experimental precision should be included. Were replicates analysed?**

We added a table in the Supplementary File, showing the experimental precision for each element based on the standard deviations of duplicate analyses. We also added a reference to this table in the manuscript. **[Lines 147-148]**

**Lines 43-44: Provide a reason why climate change might increase acid release from AS Soils.**

We added the explanation in the manuscript. Climate change is predicted to result in increasing precipitation and river discharges during winter, and increasing temperatures and evapotranspiration during summer, which is likely to enhance drying and oxidation of sulphides in AS soils. **[Lines 44-48]**

**Lines 59-67: I think the importance of understanding contaminant dynamics should be more explicit / clear – it is a valuable contribution of this research.**

We generally agree with this comment. However, this whole paragraph is about physical drivers of sediment and associated metal dynamics in the study area. We think it is already rather complete in the context of this manuscript, and are not sure what kind of information Referee would like to be added.

**Line 108: . . . cool and dark conditions - include the approximate temperature of storage, as well as time until laboratory analysis. Where/how were the samples stored in the shore-based laboratory?**

We added the temperature of storage, and the time of storage until laboratory analysis (a few months). The conditions were similar both in the ship and laboratory cold room. **[Lines 114-115]**

**Line 171: the grain-size distributions are described as poorly sorted but from a narrow grain-size range. This sounds contradictory (poor sorting implies wide grain-size range) however I think the intention is that the median grain-sizes exhibit low variability. Please clarify, and consider including the range of sorting measurements.**

That is another good point. Our intent was to state that the studied sediments are poorly sorted with rather uniform grain size distributions throughout the cores. This is now clarified in the manuscript. We also added the IQR and median of geometric sorting statistic in the manuscript. **[Lines 191-195]**

**Line 334: precipitation does not only occur 'out to sea' - explicitly identify neutralization, which can coincide with reaching seawaters.**

We added a more detailed explanation of acidic river water neutralization and consequent precipitation and deposition of Fe and Al at river mouths, and the potential complexation of other trace metals with organic matter. **[Lines 390-394]**

**Technical Comments:**

**The term contents is regularly used, when perhaps concentration would be more explicit and clear. Consider amending.**

We have used "content" for solid materials such as sediment that are measured by mass per unit mass, whereas "concentration" is used for liquids that are measured by mass per unit volume. This follows the recommendation by Flemming and Delafontaine, 2000, Continental Shelf Research.

**Line 12: which is the recipient system of the Laihianjoki and Sulvanjoki . . .**

Changed as suggested by Referee. **[Line 12]**

**Line 14: . . . landscape. Metal deposition has remained at high levels since . . .**

Changed as suggested by Referee. **[Line 14]**

**Line 26: . . . low pH conditions in pore- and surface-waters . . .**

Changed as suggested by Referee. **[Line 27]**

**Lines 28-30: Due to the uncertainty, perhaps simplify these two sentences into one.**

Done. **[Lines 30-31]**

**Line 54: deteriorative***

Changed as suggested by Referee. **[Line 58]**

**Line 63: in the long term**

Changed as suggested by Referee. **[Line 68]**

**Line 111: Define 'fresh'.**

We replaced "fresh" by "untreated" **[Line 118]**

**Line 118: . . . classified as those . . .**

Changed as suggested by Referee. **[Line 125]**

**Line 128: . . . dissolved in 1 M HNO3. . .**

Changed as suggested by Referee. **[Line 135]**

**Line 130: Instead identify which element was analysed by which technique inside brackets.**

We added a sentence which states which elements were analysed by which technique (ICP-MS or ICP-OES). **[Lines 137-139]**

**Line 168: . . . in the R software environment.**

Corrected as suggested by Referee. **[Line 172]**

**Line 195: Mn contents are low except for a strong increase**

Changed as suggested by Referee. **[Line 218]**

**Line 241: the pattern of decreasing metal contents with increasing distance**

Changed as suggested by Referee. **[Lines 264-265]**

**Line 242: The median metal contents . . .**

Changed as suggested by Referee. **[Line 265]**

**Line 251: Reword this sentence as it implies uniformity both laterally and vertically, which you go on to clarify is not the case.**

We rephrased the sentence to highlight that the median values are similar in the upper and lower core sections, and that there is little change with distance from the river mouths. **[Line 276-278]**

**Lines 287-300 – here, and some other sections of the discussion, are moderately convoluted with data, impacting readability. Some of this could be shifted into the results section, or just simplified.**

We admit that it is not easy to write about how our measured element contents compare with those measured by others in the nearby areas in an engaging way. We have tried to streamline the text as we best could.

**Line 307: begin to decrease**

Changed as suggested by Referee. **[Line 334]**

**Line 312: as a result of, for example, . . . (the abbreviation here breaks readability)**

Changed as suggested by Referee. **[Line 339]**

**Line 312-313: rather than a decrease**

Changed as suggested by Referee. **[Line 340]**

**Lines 316-323 – I do not think this paragraph is necessary. It is mostly already explained earlier in the manuscript.**

We think this paragraph is necessary because previous workers have ignored the characteristic feature of this area, which is that present day sediment deposition is restricted to small patches only. We highlight our case with an example, which is not done elsewhere in the manuscript.

**Line 401: ecotoxicological***

Changed as suggested by Referee. **[Line 460]**

This ends the comments by Referee #1
* * *
**Referee #2:**

This is a carefully prepared manuscript on trace metals from acid sulphate soils in sediments from the Baltic Sea. I have only a few minor comments, see below.

We thank Referee #2 for the encouraging words and useful suggestions on how to improve the manuscript.

Line 11. I would suggest to focus on the sites not the cores: "in sediments at 9 sites in the Kvarken Archipelago"

Changed as suggested by Referee. **[Line 11]**

Line 14. Suggested change: "a high level"

Changed as suggested by Referee. **[Line 14-15]**

Line 18. Suggested change: "in the same sediment"

Changed as suggested by Referee. **[Line 19]**

Line 29. Is it relevant to mention that the soils are currently being mapped? I would propose to combine the two last sentences of this paragraph: "In Europe, the largest occurrences of AS soils are probably found in Finland where current estimates point to AS soils occupying an area in the order of 1 million ha (Anton Boman, personal communication)."

Changed as suggested by Referee. **[Line 30-31]**

Line 33. This sounds as if the brackish phase started in the Gulf of Bothnia, whereas the salt water entered from the south. This could be solved by saying: ". . .phase, which, in the Gulf of Bothnia, began ca. 7000 years ago" or something similar.

Changed as suggested by Referee. **[Line 34]**

Line 36: suggested change: "a significant lowering"

Changed as suggested by Referee. **[Line 37]**

Line 39: suggested change: "extremely acidic"

Changed as suggested by Referee. **[Line 40]**

Line 42: suggested change: "for biodiversity"

Changed as suggested by Referee. **[Line 43]**

**Line 58. suggested change: "the distribution pattern of the metals"**

Changed as suggested by Referee. **[Line 61]**

**Line 62. What does "expanding dredging" mean? Can you rephrase?**

Replaced by "more extensive dredging". **[Line 68]**

**Line 69. How is the distance defined here? The distance to the river mouth?**

Rephrased the sentence. It now reads "…with distance from the mouths of rivers…" **[Line 75]**

**Line 71. "from the Laihianjoki . . ." Why is river in capitals here? Is it part of the name?**

Changed "river" to lower case here and throughout the manuscript. **[Line 77]**

**Line 93. Salinity has no units (it is defined relative to the conductivity of KCl), so PSU is not necessary here.**

Removed "PSU". **[Line 99]**

**Line 95. suggested change: "a thermocline"**

Changed as suggested by Referee. **[Line 101]**

**Line 100. suggested change: "the summers of 2016-2018"**

Changed as suggested by Referee. **[Line 106]**

**Line 108. suggested change: "in the cold and dark"**

Changed as suggested by Referee. **[Line 114]**

**Line 126. Why was a sieve fraction used for this analysis? Since this treatment leads to higher element concentrations, comparison to other studies may not be not directly possible, unless the fraction >63 um was small. That only the fraction <63 um was analysed needs to be mentioned in the captions of figures 2 and 5.**

The use of the sieved fraction <63 µm for multielement analyses is a common practice in geochemical analysis because cations generally are adsorbed onto fine particles. The studied sediments are fine grained,

with the median grain sizes ranging between 2 and 3 μm. Only one sample had more than 1% grains larger than 63 μm (6.7%). Because of the insignificant contribution of the larger grains in our samples, the analysis of sieved fraction is not considered to have significantly impacted our results or conclusions.

We added to the captions of Figures 2 and 5 mentions that the concentrations are for the <63 μm grain size fraction as suggested by Referee.

**Line 128. Suggested change "in HNO3"**

Changed as suggested by Referee. **[Line 135]**

**Line 170. I suggest to include the results of the grain size analysis in a supplement.**

We have added median grain size diameters and the shares of the 2–6 μm grain size class in figures for sediment cores where available in the electronic Supplementary File.

Grain size data will be published in the PANGAEA online archive, and a citation to the dataset will be updated in the manuscript upon acceptance for publication.

**Line 191: Suggested change "similar to that of other metals"**

Changed as suggested by Referee. **[Line 214]**

**Line 195: Suggesetd change: "except for a strong"**

Changed as suggested by Referee. **[Line 218]**

**Line 344-345. Here the authors write: "Also the nutrients C and N have strong positive correlations with the same grain-size range, which indicates that the metals are associated with organic particles". This is too strongly formulated. Correlation does not imply causation. I would suggest to replace "indicates" by "suggests"**

**Line 418-420. See the previous point. "The strong association of the metals and nutrients to sediment grains of the same size range (2–6 μm) indicates that the transformation of dissolved organic matter and metals to metal-organic aggregates at the river mouths is the key mechanism of seaward trace metal transport." Indicates should be replaced by "suggests"**

In both cases, replaced "indicates" by "suggests" as suggested by Referee. **[Lines 388 and 480]**

This ends the comments by Referee #2
* * *
Dr. Anders Widerlund:

**SYNOPSIS AND GENERAL COMMENTS**

**Synopsis** The manuscript investigates the spatial and temporal distribution of metals in nine sediment cores from the Kvarken Archipelago, which is affected by drainage of acid sulphate soils (ASS) in western Finland. The manuscript assesses the trace metal distribution in Gulf of Bothnia sediments with increasing distance from river mouths. The focus is on sediments deposited before and after the intensive artificial drainage of ASS that began in the 1960s. Focus is also on metals that are leached from ASS (Al, Cd, Co, Cu, La, Mn, Ni, and Zn). There is a lack of investigations regarding metal transport from ASS and through the estuarine zone of rivers in the Gulf of Bothnia, and the topic is well in line with the Aims and Scope of Biogeosciences.

**Novelty and scientific merit / significance** The study does not present any new methods or novel approaches in the interpretation of data. Although the study primarily may be important from a local point of view, results on the transport of metals from ASS through the coastal zone apparently have not been assessed previously. The study thus has a scientific merit, and is of general interest to researchers studying metal release from ASS.

**Methodology and quality of work Sampling and measurements Sampling and analytical methods used in the study are suitable and appropriate.**

We thank Dr. Widerlund for constructive and useful suggestions on how to improve the manuscript.

**For metals, the analytical accuracy is reported relative to CRMs and in-house standards. The authors should also report the analytical precision for the metals. This could be critical for interpretation of some of the metal profiles shown in Fig. 2.**

We added a table in the Supplementary File, showing the experimental precision for each element based on the standard deviations of duplicate analyses. As it can be seen in that table, the standard deviations are generally in the 1% range of the typical measured metal contents (10% for cadmium), implying sufficient precision to support our conclusions. We also added a reference to this table in the manuscript. **[Lines 147-148]**

**Aim, interpretations and conclusions** Define the aim/objective more clearly. Was the aim to assess the metal distribution in sediments with increasing distance from rivers? This is what the interpretations and conclusions are linked to.

We have now clarified the statement of research aims in the final paragraph of Introduction. We also added a reference to this table in the manuscript. **[Line 74]**

**In general the interpretations are supported by the results. However, the manuscript should briefly state the arguments supporting that the metal increase from the 1960s is really caused by leaching from ASS, and not from industrial sources. These arguments are probably reported in some of the cited references (e.g. Nordmyr et al), but could be briefly mentioned in the manuscript to strengthen the conclusions.**

These arguments were already provided in the final paragraph of Introduction. However, it seems that this was not sufficient, and we have added a similar statement in the first paragraph of Discussion to help the reader. **[Lines 272-274]**

**Readability and language  The manuscript is well written and easy to read, and the abstract accurately reflects the content of the manuscript.**

**SPECIFIC COMMENTS**

**Line Comment 57 Should read: The metal distribution . . .**

Corrected this by "The distribution pattern of the metals…" as suggested by Referee #2. **[Line 61]**

**134 I cannot se that Hg is discussed in the manuscript, so it can be removed from the methods section.**

It is true that Hg is not discussed in this manuscript. However, Hg is included in the multielement dataset that will be published in PANGAEA upon the acceptance of this manuscript for publication. We have therefore decided to keep the Hg analysis method description for the case of future studies – and it only takes a couple of sentences in the manuscript.

**247–249 Lines can be deleted. This is already mentioned in the Introduction.**

We modified this section and added a statement regarding the sourcing of the studied metals from AS soils and not from industry as Dr. Widerlund suggested in his second last General comment. **[Lines 272-274]**

**345 The fact that Cd, Co, Cu, Ni, Zn correlate with the same grain sizes as C and N does not automatically imply a causal correlation. This may well be the case, but I suggest using the term "suggest" instead of "indicate".**

Done. Replaced "indicates" by "suggests". **[Line 388]**

**Supplementary file  Title in Supplement should agree with title of the manuscript ("trace"and "Baltic Sea" are missing)**

Done. Corrected the Supplement title as suggested by Dr. Widerlund.

**Figures  Figure 5: Show "contained dark red dots" in another colour. They are now very difficult to see. This also applies to the figures (maps) in the Supplement.**

We changed a bit the color of the "dark red dots" in Fig. 5 and Supplement. We also explored various color combinations but did not manage to find anything that would be significantly better that what is already used.

**Tables Table 1: Report the sedimentation rate (cm/year) for each core**

Done. We added sedimentation rates for each core in Table 1.

**References Line 323: Should Cook et al. be 1997 or 2000?**

Thank you for pointing this out. The reference should indeed be 1997. This is now corrected.

This ends the comments by Dr. Widerlund, and our responses to the peer-review comments.

Kind regards on behalf of all co-authors,

Joonas Virtasalo, Geological Survey of Finland

[revised manuscript text omitted]